Corrected: Author correction

# Checkpoint blockade immunotherapy reshapes the high-dimensional phenotypic heterogeneity of murine intratumoural neoantigen-specific CD8+ T cells

M. Fehlings [1], Y. Simoni[1], H.L. Penny[1], E. Becht[1], C.Y. Loh[1], M.M. Gubin[2], J.P. Ward[2], S.C. Wong [1], R.D. Schreiber[2] & E.W. Newell [1]

The analysis of neoantigen-specific CD8+ T cells in tumour-bearing individuals is challenging due to the small pool of tumour antigen-specific T cells. Here we show that mass cytometry with multiplex combinatorial tetramer staining can identify and characterize neoantigen-specific CD8+ T cells in mice bearing T3 methylcholanthrene-induced sarcomas that are susceptible to checkpoint blockade immunotherapy. Among 81 candidate antigens tested, we identify T cells restricted to two known neoantigens simultaneously in tumours, spleens and lymph nodes in tumour-bearing mice. High-dimensional phenotypic profiling reveals that antigen-specific, tumour-infiltrating T cells are highly heterogeneous. We further show that neoantigen-specific T cells display a different phenotypic profile in mice treated with anti-CTLA-4 or anti-PD-1 immunotherapy, whereas their peripheral counterparts are not affected by the treatments. Our results provide insights into the nature of neoantigen-specific T cells and the effects of checkpoint blockade immunotherapy.

[1] Agency for Science, Technology and Research (A*STAR), Singapore Immunology Network (SIgN), 8 A Biomedical Grove, Singapore 138648, Singapore. [2] Department of Pathology and Immunology, School of Medicine, Washington University in St. Louis, 660 South Euclid Avenue, St. Louis, MO 63110, USA. Correspondence and requests for materials should be addressed to E.W.N. (email: evan_newell@immunol.a-star.edu.sg)

The importance of CD8[+] cytotoxic T lymphocytes in anti-tumour responses is well established but has come under intense scrutiny given advances in our understanding of the basic principles governing spontaneous anti-tumour responses in mice and the successes of various cancer immunotherapy trials in humans. To combat outgrowth of tumours, CD8[+] T cells detect tumour antigens that are displayed in the context of major histocompatibility complex class I (MHC-I) molecules on the surface of transformed cells. In addition to tumour-associated self-antigens and cancer germline antigens, tumour-specific mutant antigens (neoantigens), arising from carcinogen exposure or other causes of genomic mutations, represent a third major class of antigens that are expressed by cancer cells (reviewed in refs [1,2]). Studies in mice showed that tumour neoantigens can be rapidly identified using genomic and bioinformatic approaches[3,4] and can be used in personalized vaccines to effectively eliminate growing cancers in mice[5,6]. Subsequent human studies revealed that tumour-specific immune responses can also be boosted or induced using similar neoantigen-based cancer vaccine approaches[7,8]. Previously we (M.M.G, J.P.W. and R.D.S.) used immunogenomic approaches to identify two immunodominant neoantigens, mutant Lama4 (mLama4) and mutant Alg8 (mAlg8), in T3 methylcholanthrene (MCA)-induced sarcoma cells. We showed that these epitopes render mice bearing progressively growing tumours susceptible to tumour rejection following treatment with anti-CTLA-4 and/or anti-PD-1. This study demonstrated that neoantigens are the favoured targets of T cells reinvigorated by checkpoint blockade therapy, that vaccines generated with immunodominant neoantigens are as effective as checkpoint blockade in inducing therapeutic tumour rejection, and that tumour neoantigen-specific T cells display distinct transcriptomic signatures that reflect the type of immunotherapy applied to the tumour-bearing host (i.e., control monoclonal antibody (mAb) (exhausted CD8[+] T cells), anti-PD-1 (change in T-cell metabolism), anti-CTLA-4 (increased priming/proliferation) or the combination of anti-PD-1 and anti-CTLA-4 (increased effector function))[5].

In humans, CTLA-4 blockade results in an enhanced neoantigen-specific T-cell response[9] and broadened melanoma antigen repertoire[10]. Other studies demonstrated a correlation between the benefits of checkpoint blockade immunotherapy and the mutational burden in patients with melanoma and non-small cell lung cancer[11–13], and showed that patients with tumours enriched for clonal neoantigens have increased sensitivity to anti-PD-1/anti-CTLA-4 immunotherapy[14]. As a result, neoantigens are currently considered promising targets for personalized cancer immunotherapy[1].

Although in silico pipelines exist that are capable of successfully predicting non-synonymous mutations that can give rise to tumour-specific neoantigens[2,15], it is not clear how accurate these methods are, given that T-cell epitope usage can be influenced by many factors[16]. Mass cytometry (a.k.a. cytometry by time of flight, CyTOF [17–19]) in conjunction with peptide-MHC tetramer staining[5,15,20–22] has been shown to facilitate broad MHC-I epitope mapping, with a theoretical possibility of simultaneously assessing >1,000 T-cell antigen specificities with high sensitivity for rare antigen-specific T cells and concurrent in-depth characterization of these cells at the single-cell level[23].

Here we employ the full capacity of mass cytometry through the use of combinatorial tetramer staining together with cellular barcoding and high dimensional cellular phenotypic analysis to assess T cells targeting 81 different candidate tumour antigens in mice bearing a progressively growing MCA-induced sarcoma that is susceptible to checkpoint blockade immunotherapy[5]. This allows us to identify neoantigen-specific CD8[+] T cells and to characterize such cells simultaneously in tumours, spleens,

draining- and non-draining lymph nodes from tumour-bearing hosts. By using high-performance dimensional reduction methodology[24–27], we further profile neoantigen-specific, tumour-infiltrating CD8[+] T cells and assess the effects of anti-CTLA-4 and anti-PD-1 therapy on these cells and their peripheral counterparts.

## Results

**Identification of neoantigen-specific T cells**. To identify neoantigen-specific CD8[+] T cells in tumours as well as in peripheral tissues (i.e., spleens, draining and non-draining lymph nodes) of MCA sarcoma-bearing mice by mass cytometry, we set up a three metal combinatorial tetramer staining approach as described previously[23]. In addition to the dominant d42m1-T3 MCA-induced sarcoma mutant tumour epitopes mLama4 and mAlg8, we (M.M.G., J.P.W. and R.D.S.) previously reported to be expressed in T3, we included another set of 79 H-2K[b]-restricted predicted tumour epitope candidates (Fig. 1a and Supplementary Table 1)[5]. Single-cell suspensions from tumours, spleens, draining and non-draining lymph nodes were obtained 12 days (the time point previously reported for peak values of antigen-specific tumour-infiltrating lymphocytes (TILs) before tumour rejection[5]) after tumour cell inoculation and probed simultaneously for all 81 potential T-cell specificities, while staining with 28 different antibodies for the further identification and characterization of CD8[+] T cells (Supplementary Fig. 1B and Supplementary Table 2). Subsequent cellular barcoding facilitated the simultaneous acquisition of the cells derived from each tissue compartment[28].

For the identification of antigen-specific, tetramer triple-positive cells, we gated on live immune cells (cisplatin[−], DNA[+] CD45[+]), excluded B cells (CD19[+]) and selected TCR-beta[+], CD90[+], CD8[+] and CD4[−] cells (Supplementary Fig. 1A), and used an automated combinatorial peptide-MHC gating strategy[23], which was further confirmed by manual standard biaxial gating (Fig. 1b). Antigen specificity was defined by a standardized cutoff of >0.15 % of total CD8[+] T cells.

Consistent with previously published data[5], we identified substantial numbers of CD8[+] T cells restricted to two prevailing mutant tumour epitopes, mLama4 ($10.2 \pm 3.2\%$ as the mean $\pm$ SEM of five experiments) and mAlg8 ($9.9 \pm 3.8\%$ as the mean $\pm$ SEM of five experiments), infiltrating the tumours of tumour-bearing mice. In addition, we were able to detect CD8[+] T cells reacting with tetramers specific for both epitopes in spleens (mLama4: $0.8 \pm 0.2\%$; mAlg8 $0.6 \pm 0.2\%$ as the mean $\pm$ SEM of five experiments), draining lymph nodes (mLama4: $1 \pm 0.2\%$; mAlg8 $0.3 \pm 0.1\%$ as the mean $\pm$ SEM of five experiments), as well as non-draining lymph nodes (mLama4: $0.4 \pm 0.1\%$; mAlg8 $0.2 \pm 0.03\%$ as the mean $\pm$ SEM of five experiments) from the same group of animals. We did not track CD8[+] T cells restricted to any of the other 79 potential tumour H-2K[b]-restricted mutant epitope candidates (Fig. 1a) or controls (SIINFEKL) consistently across the different types of tissues. Moreover, none of the predicted epitope candidates was found in spleens or lymph nodes from wild-type non-tumour-bearing animals.

We next interrogated the phenotypes of T cells from each tissue by summarizing the expression profiles of molecules associated with CD8[+] T-cell differentiation (CD62L and CD44)[29] activation/stimulation (ICOS)[30], recruitment/trafficking (CXCR-3)[31] and exhaustion/dysfunction (Tim-3 and PD-1)[32,33]. We observed numerous phenotypic variations in the CD8[+] T-cell populations across tissues, which validated our antibody staining. For instance, CD44 is upregulated on antigen-experienced cells after activation[34] and we detected the highest frequencies of

CD44-positive cells in tumours. In contrast, we found higher numbers of T cells expressing CD62L and CXCR3 in the periphery, the latter which has recently been shown to be required for CD8[+] T-cell trafficking towards melanomas in vivo[35] (Fig. 1c). In contrast to their peripheral counterparts, large numbers of tetramer positive cells infiltrating the tumours also expressed PD-1 and Tim-3, and these markers could only be identified on a very low percentage of antigen-specific T cells in the periphery of tumour-bearing animals (Fig. 1c, d). These data demonstrate that mass cytometry together with a combinatorial tetramer staining approach can be used to comprehensively screen for and to phenotypically characterize CD8[+] T cells

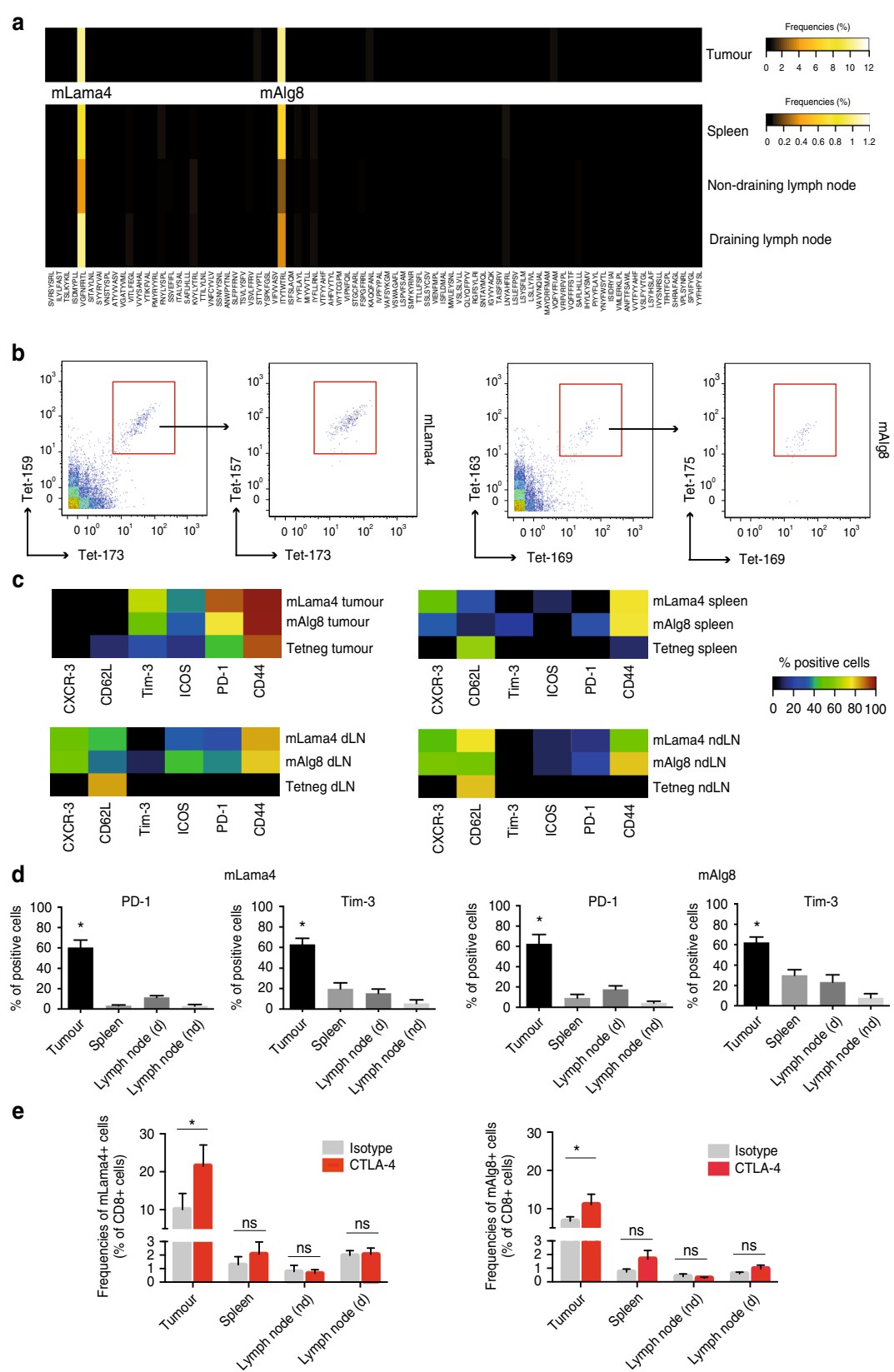

targeting a broad range of predicted epitope candidates simultaneously across multiple tissues.

**Effects of anti-CTLA-4 immunotherapy on T-cell frequencies**. Administration of anti-CTLA-4 antibodies was demonstrated to result in a CD8[+] T-cell-dependent tumour rejection and treatment doubled the numbers of mLama4- and mAlg8-specific T cells infiltrating the tumours of MCA sarcoma-bearing mice[5]. This prompted us to assess whether CTLA-4 blocking also affects the magnitude of those antigen-specific CD8[+] T cells in spleens, draining and non-draining lymph nodes of tumour-bearing mice. Moreover, as anti-CTLA-4 treatment has recently been shown to broaden the T-cell repertoire in melanoma patients[10], we wanted to assess whether CTLA-4-blocking primes and gives rise to T-cell responses against predicted antigen candidates that are not present in untreated tumour-bearing animals.

We detected a significant increase in the magnitude of mLama4- (~2-fold) and mAlg8- (~1.5-fold) specific T-cell responses in tumours from mice that received anti-CTLA-4 mAbs (Fig. 1e). The frequencies of T cells restricted to these immunodominant antigens found in the peripheral compartments, however, remained unaltered and we did not observe significant differences in frequencies of such T cells in the spleens or lymph nodes of tumour-bearing mice undergoing immunotherapy (Fig. 1e). Among all epitopes tested, we were not able to identify novel antigen-specificities (>0.15%) both in tumours or in the peripheral circulation of anti-CTLA-4-treated mice that were not also seen in isotype mAb-treated mice. These results suggest that anti-CTLA-4 treatment induces increased proliferation of neoantigen-specific T cells selectively in the tumour.

**In-depth profiling of antigen-specific TILs**. We next interrogated phenotypic alterations of mLama4- and mAlg8-specific T cells that were related to anti-CTLA-4 treatment. Although phenotypic and functional changes of antigen-specific TILs in response to immunotherapy have previously been described, it was either by assessing the expression of single marker molecules or by assessing gene expression profiles of sorted cell populations[5]. Being able to simultaneously interrogate 28 relevant surface molecules at the single-cell level by using the remaining antibodies of the tetramer staining panel, we were able to deeply profile these cells and analysed their phenotypes by applying the t-distributed stochastic neighbour embedding (t-SNE) algorithm[24–26]. By mapping cells with similar phenotypes to nearby points into a two-dimensional space, t-SNE disentangles distinct cellular subsets and reduces the high dimensionality of the mass cytometry data into two dimensions, while retaining the overall cellular relationships. We combined the data acquired for mLama4- and mAlg8-specific TILs from tumour-bearing mice undergoing anti-CTLA-4 treatment together with data acquired from isotype mAb-treated mice to produce an overall map of all

observed phenotypic profiles of tumour-specific TILs. Using these criteria and the resulting two-dimensional t-SNE dot plot, we were able to visually detect several distinct cell clusters of all possible phenotypes observed from tumour-specific T cells derived from tumours (Fig. 2a). The phenotypes of these clusters, in terms of median expression levels of each marker assessed, were summarized as heat plots for each cluster (Fig. 2a). Based on these plots, we delineated ten different phenotypic clusters of antigen-specific T cells that are composed of cells specific for either neoantigen derived from anti-CTLA-4 treated and isotype control treated tumour bearing subjects (Fig. 2a).

Based on the clusters of cells identified using t-SNE analysis, a gating strategy using standard biaxial plots was devised. This allowed us to construct working definitions for each of these cell clusters (Fig. 2b). By plotting PD-1 vs. KLRG-1 we were able to identify clusters three (C3), four (C4) and five (C5) according to their high PD-1 and their absence of KLRG-1 expression. Although all three clusters were also positive for Sca-1, C3 was negative for Tim-3, and C4 as well as C5 could further be distinguished according to their expression intensities for Tim-3 and CD39. Cluster 6 (C6) and 10 (C10) constituted KLRG-1-positive cells and could further be subdivided based on their Sca-1 and Tim-3 expression profiles. PD-1 and KLRG-1-negative cells were delineated into cluster one (C1) and two (C2), as well as clusters seven (C7), eight (C8) and nine (C9) according to their Sca-1 and Tim-3 expression levels. Clusters C1 and C2 were Sca-1 negative but exhibited differential CD27 expression, whereas C7, C8 and C9 expressed Sca-1 but showed differential expression levels of Tim-3 and CD160, respectively (Fig. 2b).

Elevated expression levels of PD-1, Lag-3 and Tim-3 on mLama4 and mAlg8-specific TILs cells have previously been linked to a dysfunctional phenotype that is accompanied by severe functional deficits of these cells[5]. According to the expression intensities of these markers found on the cells within the clusters segregated by t-SNE, clusters C3, C4 and C5 appeared to comprise such dysfunctional T cells albeit to varying degrees, whereas cluster C2 did not express any of the signature markers (Fig. 2c). Notably, we also observed high expression levels of GITR—a molecule that has been shown to be involved in tumour immunity[36,37]—on all cells that were restricted to clusters C3, C4, and C5.

These results collectively demonstrate that t-SNE is able to segregate antigen-specific TILs derived from anti-CTLA-4 treated and control mice into several distinct clusters, thus revealing a remarkable variation in the phenotypes of these neoantigen-specific TILs.

**Antigen-specific TILs constitute a heterogeneous population**. To characterize the phenotypes of the antigen-specific TILs, we focused first on data from tumour-bearing mice that did not undergo CTLA-4 blockade. From this analysis, we observed that

---

**Fig. 1** Analysis of neoantigen-specific T cells in tumours and periphery. **a** Screening for CD8[+] T cells targeting 81 potential mutant peptide–MHC complexes by a combinatorial peptide–MHC tetramer staining approach identified significant numbers of CD8[+] T cells restricted to two major mutant epitopes, mutant Lama4 (mLama4) and mutant Alg8 (mAlg8), simultaneously in tumours, spleens, draining lymph nodes and non-draining lymph nodes of tumour-bearing mice. Data are average frequencies from at least three independent experiments. **b** Representative example for a triple-coded tetramer staining from the draining lymph nodes to identify antigen-specific CD8[+] T cells. T cells specific for mLama4 were identified by tetramers labelled with Gd-157, Tb-159 Di and Yb-173 elements (0.89%), whereas mAlg8-positive cells were identified by tetramers labelled with Dy-163, Tm-169 and Yb-173 elements (0.23%). **c** Representative example for phenotypes of tetramer-positive (mLama4 and mAlg8) and tetramer-negative (Tetneg) CD8[+] T cells from tumours, spleens and lymph nodes of tumour-bearing mice. **d** Percentages of mLama-4 and mAlg8-specific CD8[+] T cells expressing PD-1 and Tim-3 in tumours, spleen, draining lymph nodes (dLN) and non-draining lymph nodes (ndLN). Data are means ± SEM of five independent experiments. *$p < 0.05$ by $t$-test corrected for multiple comparisons with Holm–Sidak procedure. **e** Frequencies of mLama4-specific CD8[+] T and mAlg8-specific CD8[+] T cells from tumours, spleen, draining and non-draining lymph nodes of tumour-bearing mice treated with anti-CTLA-4 or isotype control mAbs. Data are means ± SEM of at least five independent experiments. *$p < 0.05$ by $t$-test corrected for multiple comparisons with Holm–Sidak procedure

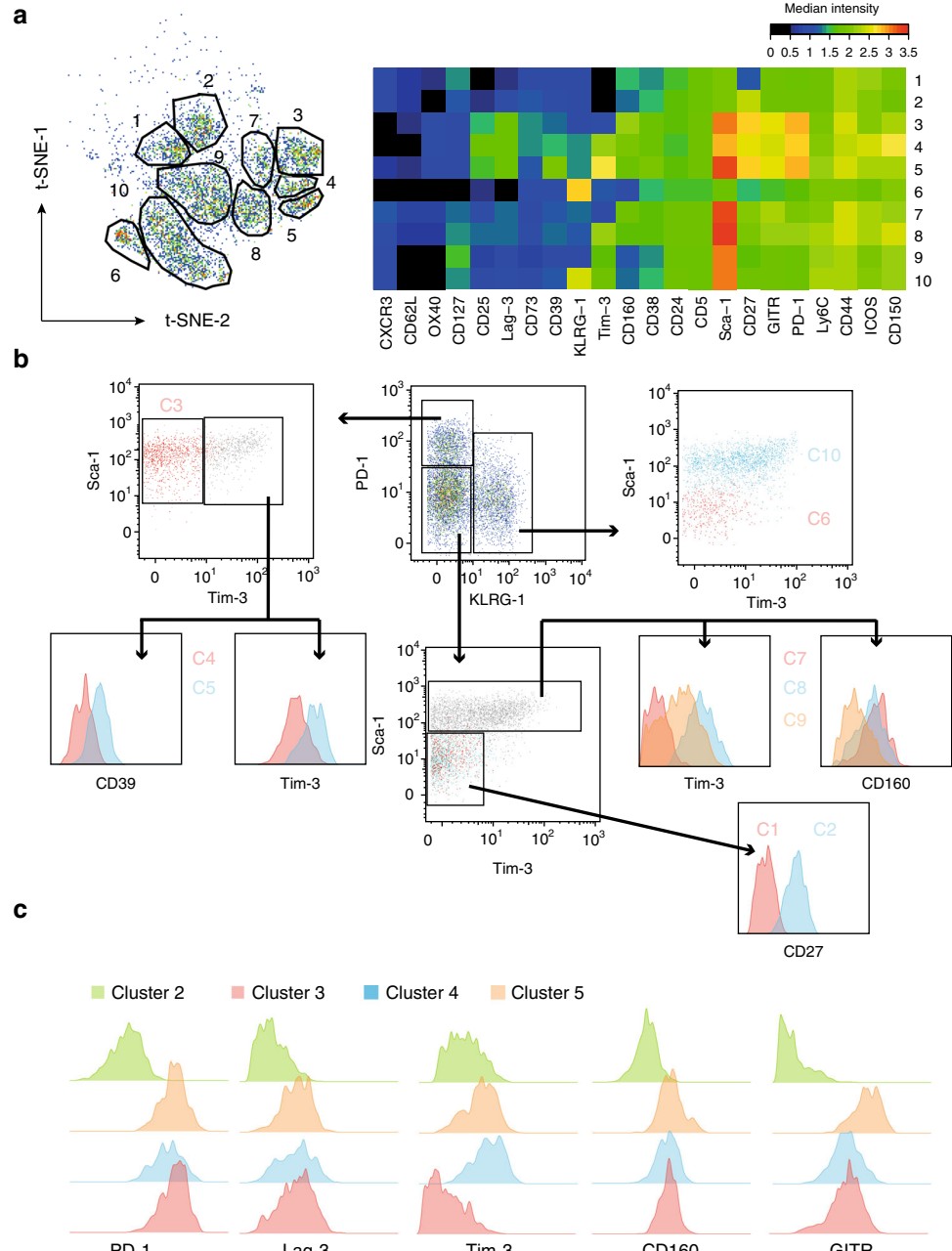

**Fig. 2** Neoantigen-specific TILs display different phenotypes. Analysis was performed on a combined data set from tumour infiltrating T cells specific for mLama4 and mAlg8 of mice treated with anti-CTLA-4 and from mice treated with control (isotype) mAbs. **a** Manual gating on antigen-specific CD8+ T-cell clusters segregated by t-SNE identified a total of 10 different clusters. The median expression intensities for each of the markers probed were plotted for all clusters observed and summarized as heat map. **b** Representation of the different phenotypic clusters delineated by t-SNE using a sequential biaxial dot plot gating strategy on signature markers characteristic for each cluster. **c** Histograms showing clusters with differential expression levels of markers associated with T-cell dysfunction (PD-1, Lag-3, Tim-3 and CD160) and GITR. Cluster 2 (C2) served as analogue

T cells specific for either antigen could be subdivided into several clusters thus representing a heterogeneous cell population amongst T cells that were restricted to a single tumour epitope (Fig. 3a). Among the 10 different clusters identified by t-SNE as described above, the majority of T cells from untreated tumour bearing mice could be found within clusters C1–C6 (*black circled*, with a frequency >10% for either one of the two antigen specificities), whereas lesser cells specific for either epitope were found to be distributed across clusters C7–C10 (*grey circled*). In particular, whereas the majority of mLama4-specific T cells could be detected within C1–C3, we found the highest frequencies of mAlg8-specific T cells to be present in C3–C6. To validate the

existence of different phenotypic clusters among tumour-specific T cells that were disentangled from the high dimensional data set by using the t-SNE algorithm, we further designed a flow cytometry panel that allowed us to delineate the predominant T-cell clusters (C1–C6) at comparable frequencies (Supplementary Fig. 2A) in untreated tumour-bearing mice according to the sorting strategy described in Fig. 2b. Owing to limitations in the number of markers available for these stainings, we were not able to clearly separate cluster four (C4) from cluster five (C5) and therefore combined the two clusters to be able to compare these stainings with the data obtained from the high-dimensional mass cytometry staining approach. Likewise, t-SNE was able to

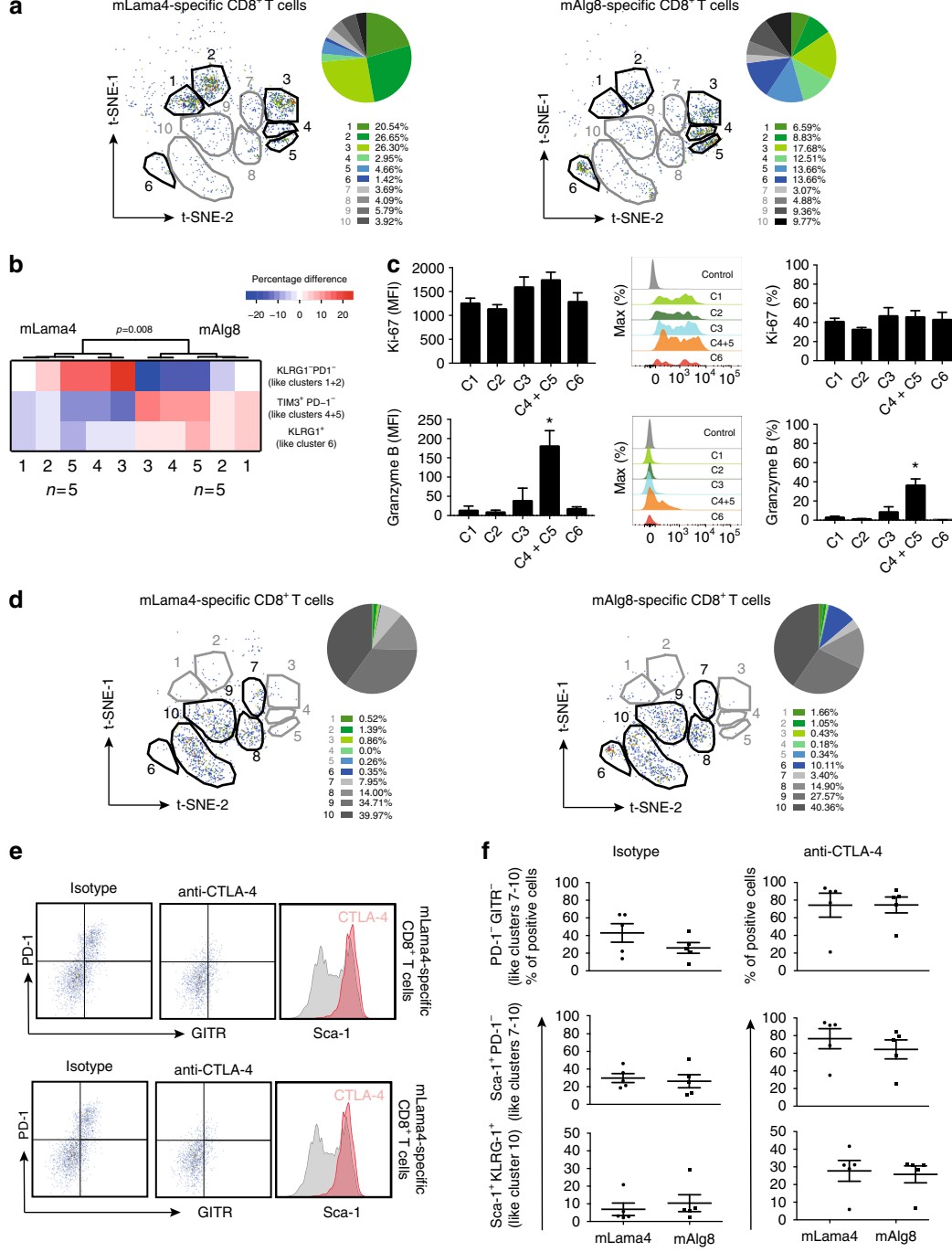

**Fig. 3** Heterogeneity of tumour-specific TILs changes following treatment. **a** The majority of antigen-specific T cells from untreated tumour-bearing mice can be identified within six clusters (C1–C6) segregated by t-SNE. mLama4 and mAlg8-specific T cells are unequally distributed with different percentages of cells found in the individual clusters (*n* = 5 mice per group). **b** The distribution pattern of mLama4 and mAlg8- specific TILs within the corresponding clusters is consistent across independent experiments. Percentage differences of mLama4- and mAlg8-specific TILs positive for the marker constellation that identify each of the clusters segregated by t-SNE (*p* = 0.008, Fisher's exact test: *n* = 5). **c** Median fluorescence intensities and percentages of cells show comparable levels of Ki-67 but differences in granzyme B expression within distinct antigen-specific T cell clusters. CD8[+] T cells from naïve spleens served as controls. Data are means ± SEM of at least three independent experiments. *$p < 0.05$ by *t*-test corrected for multiple comparisons with Holm–Sidak procedure. **d** Anti-CTLA-4 treatment shifts mLama4- and mAlg8-specific TILS from clusters C1–C5 towards clusters C7–C10 on the t-SNE plot. **e** Anti-CTLA-4 treatment of tumour bearing mice decreases the number of PD1[+] GITR[+] mLama4- and mAlg8-specific TILs and enhances Sca-1 expression on cells specific for either epitope (*n* = 5 mice per group). **f** mLama4- and mAlg8-specific T cells display higher frequencies of markers associated with cluster C7-C10 following anti-CTLA-4 treatment as compared to their control (isotype) mAbs treated counterparts and show comparable percentages amongst both antigen specificities. Data are means ± SEM of five independent experiments

segregate these clusters on a two-dimensional map albeit to a lesser degree due to the complexity restrictions associated with the flow cyometry staining panel (Supplementary Fig. 2B). Notably, by applying these stainings on tumours from individual mice, we further eliminated the possibility that the heterogeneity observed resulted from the pooling of several tumours required for the analysis by the mass cytometer.

Although T cells restricted to mLama4 or mAlg8 partly overlapped, they did not entirely cluster together. We noted remarkable differences in the cluster frequencies corresponding to the different antigen specificities. A higher percentage of mLama4-specific T cells were found within clusters C1, C2 and C3, which are represented by a KLRG-1⁻, PD-1⁻ phenotype (C1 and C2), as well as KLRG-1⁻, PD-1⁺ cells (C3). In contrast, higher percentages of mAlg8-specifc cells could be detected in clusters C4, C5 and C6, which are characterized by cells expressing both Tim-3 and PD-1 (C4 and C5), as well as cells that expressed KLRG-1 (C6) (Fig. 3a). Based on these definitions, we found these differences in phenotypic patterns between both antigen specificities observed to be consistent across independent experiments (Fig. 3b).

CTLA-4 blocking in tumour-bearing mice resulted in higher frequencies of mLama4- and mAlg8-specific T cells. To assess whether phenotypically different antigen-specific TILs derived from untreated tumour bearing mice differed in proliferation, we evaluated Ki-67 expression in cells derived from the distinct clusters by conventional flow cytometry. In addition, we evaluated the capacities of these cells to express granzyme B to assess their functional status. We found that approximately half of the cells present in each of the clusters were positive for Ki-67. However, we did not observe significant differences in the proliferation levels of these cells amongst the clusters (Fig. 3c). In contrast, whereas the majority of the clusters were characterized by a low expression of granzyme B, we detected significantly higher expression levels within cluster four and five as compared to the other clusters (Fig. 3c).

In conclusion, antigen-specific TILs constituted a heterogeneous cell population and T cells restricted to distinct mutant tumour epitopes exhibited unique phenotypic with different functional characteristics.

**CTLA-4 blocking drives T cells towards a similar phenotype.** Next, we studied the phenotypes of neoantigen-specific T cells derived from tumour-bearing mice that underwent anti-CTLA-4 checkpoint blockade immunotherapy. This analysis was striking in that cells occupying regions C1–C5 were mostly absent in tumours from these treated mice. Instead, the majority of the neoantigen-specific T cells were found in clusters C7–C10 and the frequency distributions were similar for both antigen specificities (Fig. 3d). Despite differential expression levels of markers associated with dysfunction, cells within clusters C3–C5 corresponded to cells with a PD-1⁺ GITR⁺ phenotype (refer to Fig. 2a). CTLA-4 blocking greatly reduced the frequencies of cells in clusters C3–C5 and thus nearly eliminated the cells expressing markers associated with T-cell dysfunction (e.g., PD-1 and Tim-3) (Fig. 3d). By using standard biaxial gating, we further confirmed that anti-CTLA-4 treatment resulted in increased numbers of mLama4- and mAlg8-specific TILs that display low PD-1 and low GITR expression levels (Fig. 3e). In addition, these cells exhibited a high expression of Sca-1.

We (M.M.G., J.P.W. and R.D.S.) have previously shown that besides CTLA-4 blocking, anti-PD-1 treatment rendered neoantigen-specific T cells capable of tumour rejection. To assess whether this also comes along with such a massive phenotypic alteration, we performed our high-dimensional profiling

approach on mLama4-specific TILs from mice that underwent anti-PD-1 treatment or remained untreated (isotype control). Likewise, we observed a dramatic shift of different tumour-specific T-cell clusters towards a new position on a two-dimensional t-SNE plot. Similar to changes induced by CTLA-4 blocking, we detected several alterations in the expression patterns of the different marker molecules assessed, in particular a substantial reduced expression of markers that are associated with a dysfunctional T-cell phenotype as well as an upregulation of Sca-1 (Supplementary Fig 3A).

The appearance of distinct clusters with comparable frequencies amongst mLama4- and mAlg8-specific T cells following anti-CTLA-4 mAbs therapy suggested that treatment induced neoantigen-specific TILs to acquire a similar phenotypic profile that also lacks the expression of markers associated with dysfunction. Indeed, when we compared the phenotypic characteristics of these cells with and without anti-CTLA-4 treatment across different experiments, we were able to detect higher frequencies of T cells exhibiting a PD-1⁻ and GITR⁻ phenotype regardless of their epitope restrictions (Fig. 3f). According to their phenotypic profiles, C7–10 can be classified as clusters composed of cells that express high levels of Sca-1 and low levels of PD-1 (refer to Fig. 2a). Compared with cells derived from isotype control mAb-treated tumour-bearing mice, we observed higher numbers of those cells with comparable percentages amongst the two antigen specificities (Fig. 3f). In addition, cells in C10 are associated with high KLRG-1 expression and a higher number of mLama4- and mAlg8-specific T cells with similar frequencies could be detected as compared with their isotype control mAb-treated counterparts. We found that KLRG-1 expression is also associated with C6 and we found cells displaying C6-like characteristics among T cells specific for mAlg8 after immunotherapy at comparable levels to the frequencies of mAlg8-specific T cells derived from isotype control treated tumour bearing animals.

These data show that cancer immunotherapy in this mouse model induced dramatic phenotypic alterations of neoantigen-specific T cells. Moreover, T cells with different epitope usage acquired a similar phenotype that lacked surface molecules associated with T-cell dysfunction.

**Immunotherapy affects TILs but not peripheral T cells.** To determine whether the phenotypic alterations of mLama4- and mAlg8-specific TILs also occurred in the periphery of anti-CTLA-4-treated tumour-bearing mice, we used data derived from peripheral CD8⁺ T cells to broadly analyse the resultant changes in the dominant neoantigen-specific CD8⁺ T-cell population from the two treatment groups using the aforementioned t-SNE-based dimensionality reduction approach. Owing to low frequencies of antigen-specific T cells found in each of these peripheral tissues assessed and to allow for comparisons of the phenotypes of bulk CD8⁺ T cells infiltrating the tumours, we also included tetramer-negative CD8⁺ T cells for this analysis. Notably, although the majority of tetramer-negative CD8⁺ T cells isolated from tumours were phenotypically distinct from their antigen-specific counterparts, we detected a partial overlap between tetramer-positive and negative cells on the two-dimensional t-SNE plot. We categorized a total of four clusters that showed different percentages of all TILs, and that could further be characterized according to the differential expression profiles of the marker molecules assessed (Supplementary Fig. 4A–C).

In contrast to mLama4- and mAlg8-specific TILs, very little phenotypic alterations were observed when comparing neoantigen-specific cells derived from peripheral tissues of isotype vs. anti-CLTA-4 mAb-treated mice. Whereas phenotypic changes

in antigen-specific TILs from mice treated with anti-CTLA-4 mAbs were reflected by a remarkable shift in their positions occupied on the two-dimensional t-SNE plot (Fig. 4a), we did not detect substantial differences in the cluster regions occupied by

lymph node- or spleen-derived T cells specific for either antigen when cells were obtained from mice that underwent anti-CTLA-4 treatment (Fig. 4a and Supplementary Fig. 5). Likewise, whereas we detected a remarkable change in the position of tumour-

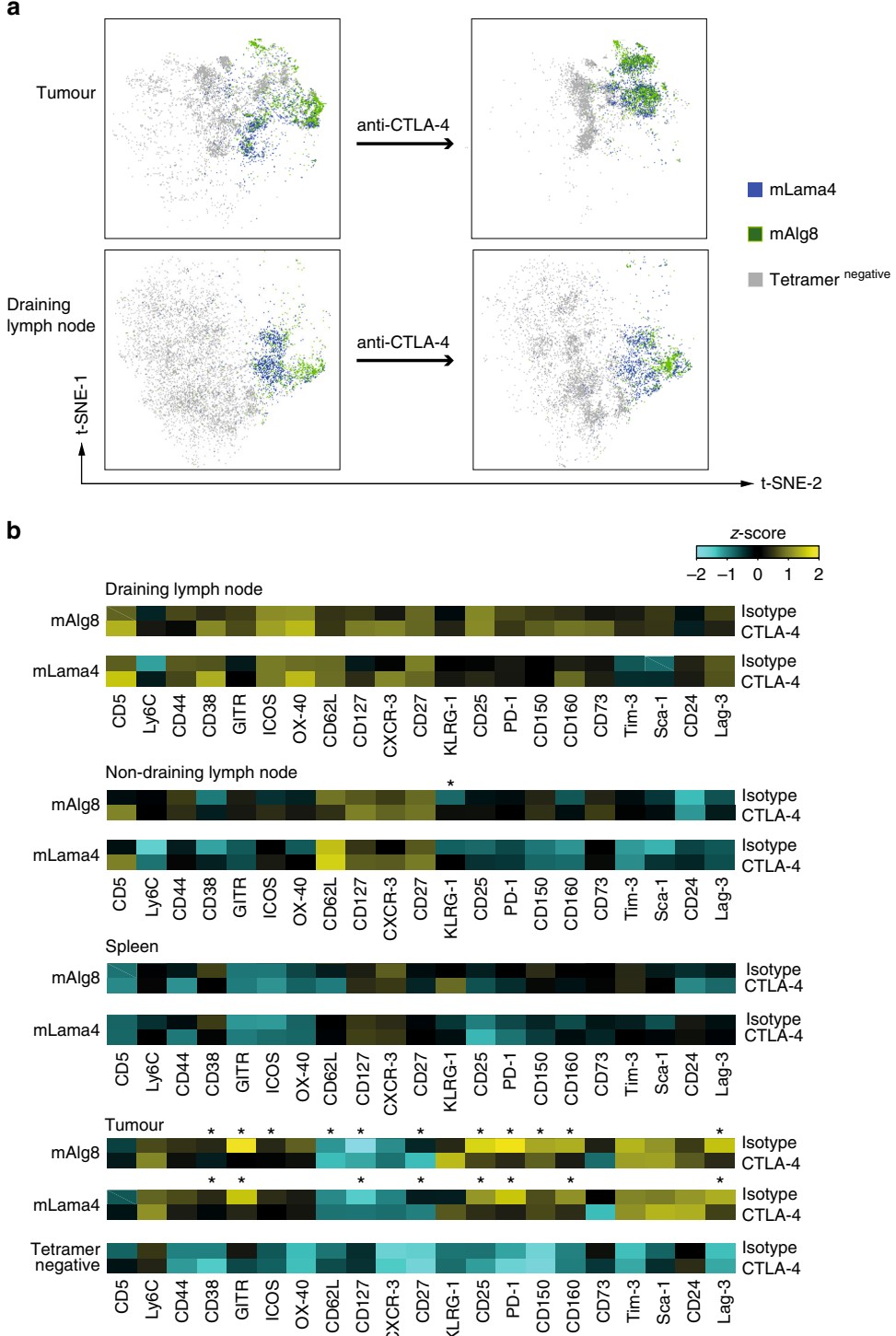

**Fig. 4** CTLA-4 blocking affects phenotypes of neoantigen-specific TILs. **a** Visualized t-SNE map highlighting the distribution of tetramer positive antigen-specific T cells and tetramer-negative cells within tumours and draining lymph nodes (as representative for the periphery). T-cell clusters specific for mLama4 or mAlg8 from the draining lymph node can be identified within similar locations on a two-dimensional t-SNE plot, irrespectively of anti-CTLA-4 or control (isotype) mAbs treatment, whereas antigen-specific CD8[+] T cells from tumours are localized within different areas following immunotherapy (n = 5 mice per group). **b** Heat plot of the relative expression of each marker molecule assessed for mAlg8- and mLama4-specific CD8[+] T cells simultaneously in lymph nodes, spleens and tumours from anti-CTLA-4 and control (isotype) mAbs treated tumour-bearing mice. Shown are the average z-scores from five independent experiments. *p < 0.05 by t-test and BH correction using a false discovery rate of 0.25

infiltrating mLama4-reactive cells in mice that underwent anti-PD-1 therapy, we did not make such observations in the lymph nodes from these mice (Supplementary Fig. 3b).

These findings suggest that the major effects of immunotherapy on T-cell phenotypes were restricted to neoantigen-specific T cells infiltrating the tumour. To further objectively compare the phenotypic profiles of tumour specific cells derived from tumour vs. periphery with and without anti-CTLA4 treatment across separate experiments, we averaged the median expression intensities from independent experiments ($n = 5$) for each of the non-lineage molecules assessed and calculated z-scores for each phenotypic marker and averaged them for each T-cell specificity as previously described[23]. We represented the data as heat plots to meaningfully assess differences in phenotypic profiles of mLama4- and mAlg8-reactive cells in the distinct tissues of mice treated with either isotype control or anti-CTLA-4 mAbs. Within the tumours of treated animals, CTLA-4 blockade gave rise to a significantly altered expression of markers associated with dysfunction (PD-1, Lag-3 and CD160), activation (CD25, GITR and CD38), as well as co-stimulation and development (CD27 and CD127) on mLama4-specific T cells. Similar findings were obtained when we interrogated the marker expression of mAlg8-restricted T cells across the different tissues (Fig. 4b). From these heat plots, it was also apparent that the phenotypes of peripheral CD8$^+$ T cells specific for either antigen, unlike neoantigen-specific TILs, were not influenced by anti-CTLA-4 immunotherapy, as no differences were observed in the expression of any of the marker molecules assessed simultaneously in the spleens and draining and non-draining lymph nodes. Likewise, we did not observe remarkable or statistically significant differences in the phenotypes of tetramer negative cells found within the tumours in response to anti-CTLA-4 treatment (Fig. 4b).

Collectively, we show that checkpoint blockade immunotherapy in this mouse model did not profoundly influence the phenotypic profiles of peripheral neoantigen-specific CD8$^+$ T cells but predominantly affected marker expression of antigen-experienced T cells present in the tumour. Although resulting from a short-term transplant model, these data suggest that neoantigen-specific TILs may be generally more susceptible to immunotherapy than their peripheral counterparts.

## Discussion

Accumulating data suggest that neoantigens are relevant targets for personalized anti-cancer therapies. Here we used mass cytometry analysis in combination with a multiplex combinatorial tetramer staining approach[23] to screen for CD8$^+$ T cells targeting a broad range of antigen candidates across tissues from mice bearing progressively growing MCA-induced sarcomas that are susceptible to checkpoint blockade immunotherapy.

Similar to our (M.M.G., J.P.W. and R.D.S.) previous findings[5], we found that neoantigen-specific TILs had a higher frequency of cells that co-expressed PD-1 and Tim-3 than tetramer-negative CD8$^+$ TILs. In addition, these cells also had higher expression levels of the PD-1 and Tim-3 markers, which is in line with a previous independent report using a MC38 colorectal cancer model where antigen-specific TILs displayed a more exhausted-like phenotype than bulk CD8$^+$ TILs[6]. In contrast to a recent CT-26 tumour study, where also lymph node-derived tumour antigen-specific T cells showed upregulated PD-1 expression[38], we only observed small numbers of PD-1-positive antigen-specific T cells in the peripheral compartments, suggesting that these cells are not being exposed to continual antigen exposure[39] or were disarmed by the suppressive tumour microenvironment[40].

We detected several tumour-infiltrating T cells that were not specific for mLama4, mAlg8 or any other epitope tested here. Interestingly, we found that their phenotypic profiles overlapped only partially with T cells that are restricted to the immunodominant epitopes. Whether these cells may be specific for other tumour-derived antigens is currently unclear. This highlights the potential utility of our approach in extending the screen for other possible antigens being targeted by these cells.

Consistent with previous findings, we detected an increase in the magnitude of mLama4- and mAlg8-specific TILs following anti-CTLA-4 immunotherapy, while the peripheral counterparts were not affected. CTLA-4 blocking has been shown to enhance priming of responsive T cells[41] and in a recent study on blood samples derived from melanoma patients it was shown that anti-CTLA-4 immunotherapy can broaden the range of antigens being targeted by the T-cell repertoire[10]. We did not detect the appearance of novel T-cell specificities against any of the epitope candidates tested here, thus suggesting a CTLA-4-dependent tumour-specific effect on the two dominant antigen-specific T cells in this model.

By using t-SNE dimensionality reduction, we provide a detailed insight into the phenotypes of antigen-specific TILs from mice treated with checkpoint blockade or control mAbs. The delineation of 10 clusters in our analysis allowed us to investigate an in-depth picture of neoantigen-specific CD8$^+$ T cells diversity. We validated that the manual cluster delineation used here was not arbitrary, as we detected a similar clustering scheme when we applied an automated clustering method (see Methods). Moreover, using manual gating we were able to further disentangle subtle differences between some of the clusters that were not revealed when applying automated clustering. This approach enabled us to broadly assess the composition of tumour-specific T cells and to describe and validate a remarkable degree of phenotypic heterogeneity within neoantigen-specific T cells in this model system.

The complexity of antigen-specific T cells has been described for viral antigens in humans and proposed to be necessary in order to achieve an adequate flexibility in anti-pathogen responses[19]. The detection of different phenotypic patterns among neoantigen-specific T cells suggests a dissimilar functionality of such T cells in the anti-tumour response. This was further corroborated by the fact that granzyme B expression was restricted to certain antigen-specific T-cell clusters. We also detected a KLRG-1$^+$ fraction among mAlg8-specific T cells that was absent on mLama4-specific T cells. KLRG-1 on terminal effector T cells has been shown to be involved in effective anti-tumour reactions[42]. Thus, mAlg8-reactive T cells might be more effective in carrying out distinct functions in anti-tumour responses as compared with their mLama4 counterparts in untreated tumour-bearing mice. Interestingly, upon anti-CTLA-4 treatment, a large number of KLRG-1$^+$ cells could be found in both mLama4- and mAlg8-reactive T cells. The preceding study detected a similar pattern of Tim-3 and Lag-3 expression among T cells specific for either mLama4 or mAlg8 antigen[5]. In contrast, we detected different expression levels of markers associated with dysfunction amongst both antigen-specific T cells. These phenotypic differences highlight the power of the t-SNE algorithm combined with high-dimensional mass cytometry to disentangle cellular subsets[25] and supports the hypothesis of a different functionality between T cells targeting different antigens in the same tumour. Interestingly, GITR was also associated with all of the dysfunctional clusters identified. GITR on regulatory T cells has recently been described as a target for improved anti-tumour responses in liver cancer[43] and a combined treatment consisting of anti-PD-1 and anti-GITR mAbs gave rise to anti-tumour immunity in mice by enhancing CD8$^+$ effector T-cell functions[44].

We (M.M.G., J.P.W. and R.D.S.) reported that checkpoint blockade immunotherapy resulted in a functional re-activation of mutant tumour antigen-specific T cells and this was accompanied by successful tumour rejection[5]. Consistent with this, we did not detect any of the pre-defined dysfunctional clusters on mLama4- and mAlg8-specfic T cells after CTLA-4 blocking. We made similar observations on mLama4-specific TILs in mice that underwent PD-1 blocking. As we cannot exclude that anti-PD-1 treatment resulted in blocking of this marker and thus making it inaccessible for the further detection by the mass cytometry staining, we excluded it from the t-SNE analysis. Nevertheless, regardless of this PD-1 invisibility by t-SNE we found that cells with differential expression levels of markers associated with dysfunction disappeared following anti-PD-1 immunotherapy.

Notably, our results also showed that antigen-specific T cells in response to anti-CTLA-4 or anti-PD-1 immunotherapy acquire a similar novel phenotypic diversity. We observed higher frequencies of Sca-1$^+$ CD8$^+$ cells in mice undergoing treatment. Sca-1 was reported to be involved in T-cell development, as well as T-cell activation and proliferation[45], and thus may reflect an activated status of neoantigen-specific T cells following checkpoint blocking.

An ability to identify tumour neoantigens is likely to become an important tool for personalized cancer immunotherapies[1]. Evolving genomic and bioinformatics approaches reveal an underappreciated diversity of potential mutant tumour epitope candidates. In order to identify therapy-relevant neoantigens and to understand the effects of immunotherapies on cells targeting such antigens, we envision the requirement to simultaneously analyse multiple tissues in a comprehensive approach similar to that described here. Our results demonstrate a proof of concept for the feasibility of probing a very large number of MHC-I restricted neoepitopes in cancer by mass cytometry with the simultaneous capacity to deeply profile neoantigen-specific T cell in the context of cancer immunotherapy. Importantly, our data show that checkpoint blockade immunotherapy resulted in remarkable phenotypic alterations specifically in neoantigen-specific TILs that are not observed in peripheral neoantigen-specific T cells. Hence, the characterization of peripheral neo-antigen-specific T cells may not always mirror responses occurring in the tumour, which should be an important consideration for the possible use of peripheral T cell response as biomarkers of therapeutic outcome.

## Methods

**Mice.** 129S6/SvEvTac (129SVE-M) male mice aged 6–10 weeks were purchased from InVivos (Singapore) and housed under specific pathogen-free conditions. All experiments and procedures were approved by the Institutional Animal Care and Use Committee (130879) of A*STAR (Biopolis, Singapore) in accordance with the guidelines of the Agri-Food and Veterinary Authority, as well as the National Advisory Committee for Laboratory Animal Research of Singapore.

**Tumour cell transplantation and antibody treatment.** MCA-induced sarcomas were generated as described previously[5]. Briefly, frozen cells from the subclonal progressor cell line d42m1-T3 (developed by the R.D. Schreiber laboratory[3]) where thawed from frozen stocks and expanded in vitro in RPMI media (Hyclone) supplemented with 10% hiFCS, 1% L-glutamine, 1% penicillin–streptomycin, 1% sodium pyruvate, 0.5% sodium bicarbonate and 0.5% 2-mercaptoethanol. Cells were collected using trypsin (Hyclone), extensively washed in Hank's balanced salt solution (Hyclone) and eventually resuspended in endotoxin-free phosphate-buffered saline (PBS; GIBCO) at a final concentration of $6.67 \times 10^6$ cells per ml. For inoculation, 150 μl of this solution was injected subcutaneously into the hind flanks of naive syngeneic recipient mice. Mice were monitored for tumour growth every second day. Tumour growth was quantified by caliper measurements and mice were killed by $CO_2$ asphyxiation before the tumour size exceeded 20 mm. For antibody therapy, mice were treated intraperitoneally with either 200 μg of anti-CTLA-4 (murine IgG2b clone 9D9, BioXcell) or anti-PD-1 (murine IgG2a clone RMP1-14, BioXcell) on days 3, 6 and 9 post-tumour transplant. For controls, mice

received either 200 μg IgG2b (clone MPC-11, BioXcell) or IgG2a (clone 2A3, BioXcell) isotype control antibodies.

**Tissue collection and processing.** Twelve days post-tumour transplantation, mice were killed by $CO_2$ asphyxiation and tumours, spleens, draining lymph nodes and non-draining lymph nodes were collected. Established solid tumours were excised from the adjacent tissue, minced and incubated with 1 mg ml$^{-1}$ collagenase D (Sigma), in complete medium for 1 h at 37 °C to make single-cell suspensions. The cells were then filtered through a 70 μm strainer (Miltenyi Biotec) and washed before removing dead cells via a dead cell removal kit (Miltenyi Biotec). Spleens were crushed, filtered and red blood cell lysis was performed by hypotonic lysis. Lymph nodes (draining and non-draining) were removed from mice, mashed and resuspended to obtain single-cell solutions. To remove aggregates, all cells were subsequently filtered through a 70-μm strainer.

**Antibody and streptavidin labelling.** Purified antibody clones and providers are listed in Supplementary Table 1. Recombinant streptavidin was expressed and refolded in house as previously described[23]. Maleimide-conjugated DN3 MAXPAR chelating polymers (Fluidigm) were loaded with heavy metal isotopes according to the manufacturer's recommendations. For antibody conjugations, 100 μg of purified antibodies lacking carrier proteins were coupled at a time with the metal-polymer structures according to the protocol provided by Fluidigm. For streptavidin conjugations, 50 μg of streptavidin at a time was conjugated to the respective metal-loaded DN3 polymer and finally diluted to a concentration of 200 μg ml$^{-1}$ for subsequent tetramerization[23,46].

**Peptides.** Mutant tumour epitopes used in the present study are based on predicted binding affinities using NetMHC 3.0 with an IC$_{50}$ of <50 nM, as well as the MHC class I epitope predictions as reported in ref. [5]. All peptides were ordered from Peptide 2.0, Inc. with a purity of at least 75% and were provided by Bob Schreiber's group from the Washington University in St. Louis, School of Medicine (St. Louis, MO). Epitope sequences used in this study are listed in Fig. 1a and Supplementary Table 2.

**Peptide-MHC loading and tetramerization.** Peptide–MHC complexes were synthesized in house as previously described[23]. Briefly, recombinant H-2K$^b$ heavy chains and human β2 microglobulin light chains were isolated from *Escherichia coli* derived inclusion bodies and refolded in the presence of a UV-cleavable peptide (SIINFEJL, Mimotopes) followed by subsequent biotinylation and purification. Specific peptide-MHC complexes were generated by UV irradiation (15 min, 365 nm) in the presence of single rescue peptides. Peptide exchange reactions were setup in 96- well plates with 100 μl of 100 μg per ml H2K$^b$ in PBS and 50 μM of rescue peptides. Exchanged peptide–MHC complexes were stored at 4 °C for a minimum of 12 h before tetramerization. For the generation of a combinatorial triple-coded tetramer staining mixture, three out of nine differently metal-labelled streptavidins were randomly combined at a time by using an automated pipetting device (TECAN) resulting in a total of 84 possible combinations. For initial screenings, we set up a second configuration carrying a completely different combination of metal-tagged streptavidin molecules as internal control. For tetramerization, these mixtures were then incubated with the single exchanged peptide–MHC complexes in the 96-well plates at a final molar ratio of 1:4 (total streptavidin:peptide–MHC). Each streptavidin combination was added in four steps with 10 min incubation at room temperature (RT) to the peptide–MHC complexes according to their afore determined coding schemes, followed by a final addition of 10 μM free biotin (Sigma). For each staining configuration the tetramerized pMHC complexes were then combined and concentrated by using a 10 kDa (Merck Millipore) cutoff filter. We finally exchanged the buffer into cytometry buffer (PBS, 2% fetal calf serum, 2 mM EDTA, 0.05% sodium azide) and filtered the tetramer cocktails before staining the cells with a concentration of ~ 500 nM per peptide–MHC molecule.

**Staining and data acquisition.** Staining of samples was carried out as described previously[23,25,27]. Briefly, three to five million cells/tissue were transferred into 96-well plates, washed once with cytometry buffer and incubated for 5 min on ice in 200 μM cisplatin (Sigma) for the discrimination of live and dead cells[47]. Cells were washed twice and each sample was stained with 50 μl of tetramer cocktail for 1 h at RT. Subsequently, cells were washed twice and then incubated with 50 μl of heavy metal-labelled antibody cocktail (Supplementary Table 1) for 30 min on ice followed by fixing the cells in 2% paraformaldehyde (Electron Microscopy Sciences) in PBS overnight or longer at 4 °C. The cells were then washed once in 1× permeabilization buffer (Biolegend) and each sample was then barcoded with a unique combination of two distinct barcodes consisting of either bromoacetamidobenzyl-EDTA (Dojindo)-linked metal barcodes (Pd-102, Pd-104, PD106 and PD108, and Pd-110) or DOTA-maleimide (Macrocyclics)-linked metal barcodes (LN-113) for 30 min on ice[25,27]. Cells were washed once, incubated in cytometry buffer for 5 min and subsequently resuspended in 250 nM iridium intercalator (Fluidigm) in 2% paraformaldehyde/PBS at RT. The cells were washed and the samples from each tissue were pooled together and adjusted to 0.5 million cells per ml $H_2O$ together

with 1% equilibration beads (EQ Four element calibration beads, Fluidigm) for acquisition on the mass cytometer[27].

**Flow cytometry staining**. H-2K$^b$ peptide-MHCI complexes with mLama4 or mAlg8 peptides were prepared by UV-induced ligand exchange as described above and tetramerized by three additions of streptavidin conjugated to either phycoerythrin or allophycocyanin (Biolegend) to achieve the final molar ratio of 1:4 (total streptavidin:peptide–MHC). Cells were stained with tetramers and Fc block (anti-CD16/32, BD Bioscience) in PBS 0.5% BSA at 4 °C for for 1 h at RT. Subsequently, cells were washed and then stained with fluorescently conjugated antibodies to PD-1 (29 F.1A12, Biolegend 1/100), KLRG1 (2F1, Biolegend 1/100), CD8 (53–6.7, Biolegend 1/100), Sca-1 (D7, Biolegend 1/100), TCR-beta (H57-597 Biolegend 1/100), CD27 (LG.7F9, Biolegend 1/100) and Tim-3 (B8.2C12, Biolegend 1/100), and Live–dead (Thermo-Fisher—L34957) at 4 °C for 15 min, followed by washing and incubation in FoxP3 fixation buffer (eBioscience) for 30 min on ice. After washing in permeabilization buffer (eBioscience), the cells were then stained with Ki-67 (11F6, Biolegend 1/100) and granzyme B (GB7, Biolegend 1/100) antibodies in permeabilization buffer for 30 min on ice. All flow cytometry experiments were performed on a FACSCalibur device (BD Biosciences) and analysed using FlowJo.

**Data analysis**. Mass cytometry data were analysed as previously described[25,27,48]. First, the signal of each parameter was normalized based on the equilibration beads added to each sample[49]. The zero values of the.fcs files were randomized by an R-script that uniformly distributes values between minus-one and zero. Each barcode combination was deconvolved manually followed by gating on live CD8$^+$ T cells using FlowJo software (Treestar, Inc.). Antigen-specific triple-tetramer-positive cells were further identified by an automated gating strategy[23] and subsequently validated by manual gating using Flowjo. For the detection of triple-tetramer-positive cells, we set a cut off at a frequency of 0.15% for all CD8$^+$ T cells, as signals became random below this threshold. For t-SNE dimensionality reduction, the cell events of all tissues were down-sampled to a maximum number of 10,000 CD8$^+$ T cells per tissue. t-SNE analysis was carried out by using an R-package[27] including the "flowCore" and "Rtsne" CRAN R packages for an efficient implementation of t-SNE via the Barnes-Hut approximations)[24,26]. For automated clustering, we performed k-means clustering of the t-SNE output, using 10 centres and 1,000 random repeats. The $\chi^2$-test was used to assess the correlation between the two grouping methods.

In R, all data were transformed using the "logicleTransform" function by using the "flowCore" package (parameters: $w = 0.25$, $t = 16,409$, $m = 4.5$, $a = 0$). We calculated the percentages and median intensity values for each marker assessed and used heat maps to represent marker expression and to identify the characteristic markers of each cluster. Pie charts and bar graphs shown in this manuscript were generated using Graphpad Prism software and heat plots were generated using custom R-scripts.

**Statistical analysis**. Statistical comparisons for continuous variables between two groups were done by Student's $t$-test with pooled variance and Holm–Sidak procedure for multiple comparisons, or Fisher's exact test for categorical variables. For more than two groups analysis of variance was used followed by Holm–Sidak's multiple comparisons test. For a given marker, we computed $Z$-scores by performing the transformation $X$ significance of $X \rightarrow (X-m)/s$, where $X$ is the expression of the marker across tetramer-positive cells, $m$ its mean and $s$ its SD[23]. Significance of $Z$-scores was analysed by $t$-tests followed by Benjamini–Hochberg multiple test correction using a false discovery rate threshold of 0.25[25,27].

**Data availability**. Data that support the findings of this study are available through the Flow Repository (http://flowrepository.org/id/FR-FCM-ZY8A). All other data are available from the authors on reasonable requests.

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

## Acknowledgements

We thank the SIgN community and all members of the Newell lab for helpful discussion and technical support. This study was funded by A-Star/SIgN core funding (E.W.N.) and the A-Star/SIgN immunomonitoring platform funding (E.W.N). This work was also supported by grants to R.D. Schreiber and subcontract to E.W. Newell from the NCI (RO1 CA190700).

## Author contributions

M.F. and E.W.N. conceived and designed the experiments, analysed and interpreted data, generated the figures and wrote the manuscript. M.F. performed all experiments. Y.S. provided technical support, analysed and interpreted data. H.L.P. initiated and guided performing animal experiments. E.B. helped with the analysis and statistics. C.Y.L., M.M.G., J.P.W., S.C.W. and R.D.S. provided materials and/or critical analytical support.

## Additional information

**Competing interests:** R.D.S. is a cofounder, stock holder and scientific advisory board member of Jounce Therapeutics and Neon Therapeutics, and a member of the scientific advisory boards of BioLegend, Constellation, Lytix and NGM. He also received research funding from Janssen and Agios. E.W.N. is a board director and shareholder of immunoSCAPE Pte. Ltd. M.F. is Director, Scientific Affairs and shareholder of immunoSCAPE Pte. Ltd. All other authors declare no competing financial interests.

