## [Peer Review File · Nature Communications]

Reviewers' comments:

Reviewer #1 (Remarks to the Author):

M. Fehlings et al. use an approach combining mass cytometry and multiplexed multimer labeling to characterize neoantigen-specific CD8 T cell responses in different tissues of tumor bearing mice upon immune checkpoint blockade therapy. This is basically a methods paper illustrating the usefulness of this combined experimental fine grained analysis to generate high content information about the dynamics of T cell responses to predicted neoantigens in an experimental chemically induced tumor model.

The algorithm used to deconvolute high dimensional phenotypic data (t-SNE) was reported and illustrated back in 2013, in work from the GP Nolan's lab. In the present manuscript the t-SNE statistical approach is used to generate two-dimensional clusters of antigen-specific CD8 T cells. It shows a rather impressive power to reduce the complex cobweb of median cell surface marker expression values into subsets of apparently distinct populations of T cells which are shifted upon anti-CTLA-4 treatment. The results display is elegant and very useful as it points to a tractable way of capturing the complexity of the changes operated in the makeup of neoantigen-specific CD8 T cells during anti-CTLA-4 therapy, which in this model is highly clinically relevant.

The manuscript is clearly written and the results are worth sharing with the scientific community because of their key relevance for the very fast moving field of cancer immunotherapy with immune checkpoint blockade. In particular, these analyses significantly expand the initial set of observations published by Gubin M et al, Nature 2014. There are in this regard, a number of specific points that need to be addressed in order to better put the present results into context with the previous report on the targeting of two dominant neoantigens by immune checkpoint blockade therapy.

1) The RNAseq analyses of neoantigen-specific CD8 T cells (anti-mLama4) harvested from TILs after anti-CTLA-4 therapy showed a distinctive gene signature pointing to increased NFAT and JAK-STAT signaling, cellular proliferation/cell cycle and activation of effector T cells (Gubin et al. 2014). It is a bit disappointing that no proliferation/cell cycle markers were included in the panel design used in this manuscript so as to assess the extent to which neoantigen specific CD8 T cells are in cell cycle, in particular which in the S-phase, by the time of analysis in the isotype and anti-CTLA-4 groups of tumor bearing mice. Inclusion of this analysis would increase the potential impact of this manuscript.

2) In the original work published by Gubin et al, the tumor model also responded to anti-PD-1 and to the anti-CTLA-4/anti-PD-1 monotherapy and combination therapy, respectively. The authors should also show the phenotypic changes that may be detected using this powerful analytic approach upon anti-PD-1 therapy. In fact, the latter therapy is nowadays far more relevant to the current clinical situation as anti-PD-1 therapy is showing activity in a surprisingly large number of different tumor types.

3) The most important concern with the present manuscript is the extent to which the clusters of T cells which can be traced back to defined constellations of cell surface markers are functionally operational subsets of T cells. However, this reviewer recognizes that careful functional characterization of these clusters (a minimum of 10 interesting clusters as defined in figure 3) is an enterprise exceeding the time needed to revise a manuscript. However, the authors ought to, at the very least, provide independent validation for the identity of these clusters. This is needed in view of the density of the information and the strict dependence on statistical treatment of the CyTOF data. Indeed, all the information presented in this manuscript depends on the processing of the CyTOF data by the t-SNE algorithm. Thus, the authors should demonstrate that the 10 clusters of CD8 T cells

demonstrated in figure 3 exist and are identifiable by performing classical multicolor flow cytometry. Here they should use panels of fluorescent antibody conjugates specific for the defining sets of cell surface markers in the multidimensional t-SNE maps.

4) Typos and minor point:

Page 7: line 15, there is some link missing, a parenthesis is also missing

Page 9: line 18, isotype (not isotope)

Page 10: line 6, increased proliferation (and not increases)

Page 16: line 14, the reference should follow the same format as in the rest of the manuscript (numbered)

In the introduction, the authors should update the part on neoantigens in clinic by discussing the paper by McGranahan ... Swanton, Science 2016 (Clonal neoantigens elicit T cell immunoreactivity and sensitivity to immune checkpoint blockade).

Reviewer #2 (Remarks to the Author):

The authors combined mass cytometry analysis with multiplex combinatorial tetramer staining to identify and characterize neoantigen-specific CD8+ T cells across tissues in mice bearing T3 methylcholanthrene (MCA)-induced sarcomas following checkpoint blockade immunotherapy. One of the major conclusions that two neoantigen-specific CD8 cells were activated with anti-CTLA4 treatment was already reported in the previous report (reference 5). The further classification of neoantigen-specific T cells using high dimensional phenotypic profiling is very interesting, but very descriptive. In overall, this manuscript is well written, but they need to characterize the biological significance of these subpopulations.

(1) A simple question is whether TCR repertoires in these 10 clusters in Figure 3A are identical, similar, or different. This is very important to address whether the affinity of TCR to the HLA-antigen complex influences to the phenotype differences. Since the numbers of cells in some clusters are small, they should characterize TCR sequences of at least major clusters, for example, clusters 1-5 for mLama4 T cells and for clusters 1-6 for mAlg8.

(2) They should also characterize the cytotoxic activity of T cells in some clusters to demonstrate the functional significance of these phenotypic differences. It is totally unclear in the difference in expression levels of cytotoxic molecules related to CD8 anti-tumor activity.

They must provide some biological aspects how the cells in individual clusters have different functional roles.

Reviewer #3 (Remarks to the Author):

1. The manuscript claims that checkpoint blockade reshapes the heterogeneity of intratumoral neoantigen specific CD8+ T cells. This claim is too broad as it focuses on CTLA-4 blockade only, there is no change in the epitopes recognized and so far only phenotypic differences are demonstrated. A claim that CTLA-4 blockade changes the phenotype of tumor-specific CD8+ T cells in one mouse model is justified.

2. The authors use a recently published murine tumor model in which two neo-epitopes were identified. The results on page 6-8 describe a screen on TIL, spleen and lymph nodes using combinatorial coded tetramers in which 79 new potential neo-epitopes as well as the two previously identified epitopes is shown. No new epitopes were identified. This seems to be expected as I can imagine that a similar type of screen has been performed to identify the known two epitopes (their ref 5). If CTLA-4 blocking would have changed this (which did not occur) than it would be justified to report this screen in the main figures but now it seems redundant and it could have been shown in one of the supplemental figures. A brief report that such a screen did not reveal any other neo-epitopes to be recognized, neither spontaneously nor after CTLA-4 blocking would have been sufficient. The method itself has been published before (their ref 23) and as such novelty is low for this part of the study.

3. The first results in this study reveal upregulated Tim-3 and PD-1 in TIL. This brings up the question why the authors chose to study the effects of CTLA-4 blocking rather than PD-1 (or Tim-3) blocking. It is of strong interest to know whether blocking these other molecules also results in a strong phenotypic change or whether the two antibodies may have complementary effects. Especially, since these CTLA-4 blocking and PD-1 blocking is tested simultaneously in the human setting. I would recommend to add such a study to the current one.

4. The results described at page 10 (Figure 2) is the actual novel part of the study, which I would consider to be the start of the results section. At page 12 the authors state that elevated expression of the immune inhibitory receptors are widely accepted to describe an exhausted or dysfunctional T cell subset. This view, however, has been challenged by many who show that such cells may actually represent recently activated T cells (e.g. Tas et al Cancer Res 2016; Zelinkyy et al J. Immunol 2011; Gros et al. JCI 2014). Indicating that the expression of these markers do not indicate exhausted/dysfunctional T cells per sé. The authors should more carefully describe their results refraining from statements about functional activity as this has not been tested in this study.

4. At page 12, the authors conclude that without CTLA-4 blocking the neo-epitope specific CD8+ T cells are among clusters 1-6 (out of 10). This is based on a cut-off of 10%. It is not clear why they used this cut-off. Scrutinizing the data reveals that the majority of mLAMA-4 specific T cells cluster in C1-C3 (>75%), while the majority of mAlq8 cluster in C3-6, C9 & C10 (>60%), based on the frequencies provided. So there is clear phenotypic heterogeneity between the two different antigen-specific CD8+ T cell populations. The authors should describe this much better rather than concluding that for both specificities they mostly fall into C1-C6. Their final conclusion on this matter (page 13) is correct.

5. It is spectacular to see how CTLA-4 blocking drives the majority of the cells into C8-10 (rather than C7-10, as there is more difference in C7 between the two groups). It not unexpected as in their earlier paper (their ref 5) the RNAseq and GSEA data set analyses already showed that CTLA-4 blocking induced functional differences associated with T cell activation. Anyway, the data indicate that CTLA-4 blocking really converges the phenotype of the two different neo-epitope specific populations. To understand the relevance of these mouse model data is it is highly recommended to complement the data set showing that the T cells with the new phenotype (C8-C10), indeed bear more functionality than the T cells in the other clusters.

6. The authors also interrogated potential phenotypic changes in the periphery of treated mice. Based on the results presented in Figure 4b the authors conclude that the changes found are only seen in tetramer (neo-epitope specific) CD8+ T cells since tetramer negative T cells do not show significant changes. In my opinion one should be careful to make such a statement as the population of tetramer-negative cells also contain a lot of naive T cells. These cells will dilute the signals provided by other memory or activated CD8+ T cells. If the authors want to make such a statement they should

provide data on unrelated but previously activated T cells (e.g. virus specific T cells) .

7. In summary, the authors used one mouse model to provide good evidence that CTLA-4 blocking alters the phenotype of tumor-specific T cells, most prominently in the tumor itself. This is highly interesting but a number of questions that are important to address remain:

- a) Are the changes observed coupled to a more functionally effective T cell?
- b) How general is this phenomenon, do the authors also observe this in a second mouse tumor model?
- c) Does this change also occur in patients treated with CTLA-4?
- d) Is this effect specific for CTLA-4 blocking or does it also occur when other checkpoint blockers are used? Alternatively, would the changes induced by different blockers complement each other? Based on the RNAseq and GSEA data set provided in the earlier study (their ref 5) one should expect that.

8. The discussion (pages 18-23) is rather lengthy and the first two pages more or less discuss the use of the combinatorial coded tetramers and the CyTOF as a tool to identify and phenotype tumor-specific T cells. As in this study no other epitopes were identified, and the method has already been discussed (their ref 23) this part is rather redundant and could be removed easily.

9. Furthermore, at page 20 the authors conclude that in their model CTLA-4 blocking did not lead to the appearance of novel T-cell specificities...suggesting that CTLA-4 acted only on pre-existing T cells. This can not be concluded as for none of the other peptides it has been shown that they function as actual epitopes. For such a statement, they should focus on other models for which it is known that epitopes are present but do not lead to a spontaneous immune response.

Reviewer #4 (Remarks to the Author):

The manuscript by Fehlings et al describes the combined use of mass cytometry and multiplex combinatorial tetramer staining to identify and characterize neoantigen-specific CTLs in sarcoma bearing mice. They tested 81 candidate antigens and discovered T cells specific for two previously identified neo-antigens. They found tumor-infiltrating T cells to be heterogeneous. anti-CTLA-4 immunotherapy drastically changed their profile.

This is a technically superbly executed set of experiments by the Newell lab. The paper is well-written and the conclusions are supported by the data they describe. There are only a few shortcomings which should be addressed:

The paper is an excellent technical proof-of-principle report on the feasibility of the method. The combination of CyTOF and the tetramer library is really clever, but was already described previously in their Nat Biotech paper from 2013. Here, the only novel thing is the use of known cancer antigens (Ref. 5, cited 14 times) compared to the previously investigated viral antigens. This is however mitigated by the technical brilliance of the work, but it would have been perhaps a good idea to extend the study to identify new antigens (unknown) and thus use additional cancer cell lines (e.g. B16).

The conclusions are based on the assumption that the t-SNE algorithm allows the definition of cell clusters. However, this algorithm is instead a visualization tool for dimensionality reduction, which per se does not cluster. The gating used in fig.2A, does not even allow a "visual segregation" of the different populations as depicted by the authors. For this reason, I recommend the use of automated-clustering algorithms (like self-organizing maps) to confirm the findings and characterize the different immune population in a more unbiased manner.

For Fig.1A is there biological control for the other tumor epitopes?

Fig.3: I think they over-interpreted the data. The effect of CTLA-4 must be compared with the overall impact on the tetramer negative subset. What happens when one compares the overall population of CD8 TILs profile before and after aCTLA-4 therapy? Most of the stratifying markers are also associated with an cell activation, and from this analysis one cannot dissect the direct effect of CTLA-4 on these subsets rather than an overall effect on the tumor microenvironment.

In Fig.4B they compared the expression of some activation/maturation/exhaustion markers on neo-antigen specific CD8 T cells. Interestingly, within the tumor environment, the heatmap depicts a tetramer negative compartment of CD8 T cells that express very low levels of each of the analyzed markers, including CD44, CD27, CD5 and exhaustion markers such as PD1. What are those cells? On the same line, they do not provide the tetramer negative profile from spleen and lymph nodes. What is the impact of anti-CTLA-4 on these cells?

Lastly, the title is misleading. I expected that they applied this to patient material. The title should/must say "mouse"!

REVIEWERS' COMMENTS:

Reviewer #1 (Remarks to the Author):

The authors have carefully revised their manuscript and added essential new data, including the extensive phenotypic characterization of neoantigen-specific T cells before and after PD-1 treatment. There are still specific concerns with the new data sets. While they show the Ki65 expression in TILs from untreated mice, the question was also concerning the relative levels of this proliferation marker after immune checkpoint blockade treatment. The pretreatment levels appear already quite high. Are they further increased upon anti-CTLA-4 and/or anti-PD-1 treatment?

The other concern is relative to the legibility of the results depicted in supplementary figure 3A. It is unclear how the two dimensional plots reflect the changes upon treatment. The figure legend should better explain the color codes and what data is before and after treatment.

Reviewer #2 (Remarks to the Author):

Although they improved the manuscript extensively, they did not perform the TCR analysis. It is very important to know the biological differences such as T cell clonality or granzyme levels in these clusters. Since TCR sequencing methods are established now, I wish the authors add these analyses.

Reviewer #3 (Remarks to the Author):

The authors have elaborately addressed my previous remarks, most of them to my satisfaction. A few concerns still remain.

1. The title still does not properly reflect the findings. As there were no changes in specificities but only phenotypical changes, the title should indicate this. This is simple by inserting the word "phenotypical" between "...high-dimensional" and "heterogeneity...".

2. I did not state that the concept and findings in this paper are not new. This remark was placed in the context of using mass cytometry for the combinatorial metal-based MHC tetramer screening in combination with other phenotypic markers. This work is well appreciated but has been demonstrated for 109 different tetramers in blood by the senior author (Newell et al. Nat Biotech 2013). Hence, the technique as well as the fact that it can be used for blood analyses is not new. Although the authors point out that this is the first report demonstrating the value of using mass cytometry for neo-antigen-specific T cells, I would consider this a stretch. There would for sure be a value when the authors had demonstrated that during therapy there were changes in the neo-antigens recognized. This was not the case. The sheer fact that now MHC tetramers with neo-antigen peptides are used should not be considered as something novel. Therefore, again I suggest to compress the description of the first part of the results section (pages 6-8).

3. At this point only one mouse model has been used. The authors stated that to generalize this phenomenon they would require another mouse model with known tumor-specific antigens and responsive to check-point blockade, but that it was not the intention of this study to identify such mouse model. This is a bit of a surprise as the authors must be aware of the MC38 mouse model which is responsive to the combination a-LAG3 Ab/a-PD1 Ab (Woo et al Cancer Res 2012, p917) and presents well-described neoepitopes to T cells (Yadav et al., Nature 2014, p572)

Reviewer #4 (Remarks to the Author):

satisfied! well done!

Please find attached a revised manuscript entitled “Checkpoint blockade immunotherapy reshapes the high-dimensional heterogeneity of murine intratumoral neoantigen-specific CD8⁺ T cells” by Michael Fehlings et al. that received generally positive comments in its first review but also required addressing some of the reviewers’ issues. We addressed these concerns and have added new data that we think significantly strengthens the paper. We have attached a point-by-point response to the reviewers’ comments. We have identified the specific changes we made in response to the comments by underlining them here and have also underlined corresponding changes in the revised manuscript. In addition, we start this letter with a short summary of the new data that we have added to the study:

- 1) In the previous version we assessed the effects of anti-CTLA-4 treatment on the phenotypes of neoantigen-specific T cells in tumors and periphery. In the current version we have now included data on the effects of anti-PD-1 treatment on the phenotypes of neoantigen-specific CD8⁺ T cells in tumors and lymph nodes of d42m1-T3 sarcoma bearing mice assessed by mass cytometry. Importantly, we show that anti-PD-1 checkpoint blockade in this mouse tumor model also results in dramatic phenotypic changes in neoantigen-specific TILs similar to those observed with anti-CTLA4 treatment whereas phenotypes of their peripheral counterparts were not affected.
- 2) In the previous version we identified several clusters of neoantigen-specific TILs from tumor bearing mice by mass cytometry and high-dimensional reduction analysis using t-SNE that displayed a cluster-specific characteristic phenotypic profile. By using conventional flow cytometry on individual tumors we now validate our previous findings and show that neoantigen-specific TILs indeed constitute a heterogeneous cell population that can be distinguished according to their specific marker profile. In addition, we included Ki-67/granzyme B antibodies in our flow cytometry panel and assessed the proliferation and functional status of such neoantigen-specific TIL subsets. We show that granzyme B expression in untreated tumor bearing mice varies in different T-cell subsets, whereas proliferation levels are comparable between antigen-specific cells derived from distinct clusters.

In sum, we believe we have addressed the criticisms/suggestions of the reviewers and thank the reviewers for critically evaluating this manuscript. We hope you agree with us that this is a strong and interesting paper and that you now find it worthy of acceptance for publication in Nature Communications.

Please let me know if you have any additional questions.

REVIEWER COMMENTS:

Reviewer #1 (Remarks to the Author):

M. Fehlings et al. use an approach combining mass cytometry and multiplexed multimer labeling to characterize neoantigen-specific CD8 T cell responses in different tissues of tumor bearing mice upon immune checkpoint blockade therapy. This is basically a methods paper illustrating the usefulness of this combined experimental fine grained analysis to generate high content information about the dynamics of T cell responses to predicted neoantigens in an experimental chemically induced tumor model.

The algorithm used to deconvolute high dimensional phenotypic data (t-SNE) was reported and illustrated back in 2013, in work from the GP Nolan's lab. In the present manuscript the t-SNE statistical approach is used to generate two-dimensional clusters of antigen-specific CD8 T cells. It shows a rather impressive power to reduce the complex cobweb of median cell surface marker expression values into subsets of apparently distinct populations of T cells which are shifted upon anti-CTLA-4 treatment. The results display is elegant and very useful as it points to a tractable way of capturing the complexity of the changes operated in the makeup of neoantigen-specific CD8 T cells during anti-CTLA-4 therapy, which in this model is highly clinically relevant.

The manuscript is clearly written and the results are worth sharing with the scientific community because of their key relevance for the very fast moving field of cancer immunotherapy with immune checkpoint blockade. In particular, these analyses significantly expand the initial set of observations published by Gubin M et al, Nature 2014. There are in this regard, a number of specific points that need to be addressed in order to better put the present results into context with the previous report on the targeting of two dominant neoantigens by immune checkpoint blockade therapy.

- We thank this reviewer for the positive comments and are pleased that our study was so well received.

- 1) The RNAseq analyses of neoantigen-specific CD8 T cells (anti-mLama4) harvested from TILs after anti-CTLA-4 therapy showed a distinctive gene signature pointing to increased NFAT and JAK-STAT signaling, cellular proliferation/cell cycle and activation of effector T cells (Gubin et al. 2014). It is a bit disappointing that no proliferation/cell cycle markers were included in the panel design used in this manuscript so as to assess the extent to which neoantigen specific CD8 T cells are in cell cycle, in particular which in the S-phase, by the time of analysis in the isotype and anti-CTLA-4 groups of tumor

bearing mice. Inclusion of this analysis would increase the potential impact of this manuscript.

- To address this important issue raised by the reviewer, we designed a flow cytometry panel including Ki-67 to assess the proliferation status of distinct neoantigen-specific TIL subsets (Figure 3C). When we analyzed the proliferation status from different T-cell clusters found in untreated tumor bearing mice, we detected that approximately half of the cells found in the clusters were positive for Ki-67. However, regardless of clear phenotypic differences amongst the T-cell clusters we did not detect remarkable differences in the proliferation levels of the cells within these different clusters (Page: 14).
- 2) In the original work published by Gubin et al, the tumor model also responded to anti-PD-1 and to the anti-CTLA-4/anti-PD-1 monotherapy and combination therapy, respectively. The authors should also show the phenotypic changes that may be detected using this powerful analytic approach upon anti-PD-1 therapy. In fact, the latter therapy is nowadays far more relevant to the current clinical situation as anti-PD-1 therapy is showing activity in a surprisingly large number of different tumor types.
- The scope of our work was to demonstrate the feasibility of a mass cytometry based multiplexed tetramer staining approach to screen for a large number of neoantigen candidates across tissues while retaining the capacity to perform in depth profiling and subset identification of neoantigen-specific cells in a tumor model that is responsive to checkpoint blockade immunotherapy. To these ends we performed our experiments by using one checkpoint inhibitor that has been shown to result in remarkable changes in this model (anti-CTLA-4, Gubin et al., 2014). Nevertheless, we agree with the reviewer on the importance of assessing tumor antigen-specific T cells upon anti-PD-1 immunotherapy and performed additional experiments with mice undergoing anti-PD-1 treatment and assessed the phenotypic changes on mLama4-neoantigen-specific TILs and their peripheral counterparts. We found that anti-PD-1 treatment induced substantial phenotypical changes on mLama4-specific T cells in the tumors and that these changes are comparable to those observed with anti-CTLA-4 treatment. Notably, we also found that these changes are restricted to TILs, since we did not detect such changes when we investigated mLama4-specific T cells in draining lymph nodes following anti-PD1 immunotherapy. (Page: 15 and 22) (Supplementary Figure 3).
- 3) The most important concern with the present manuscript is the extent to which the clusters of T cells which can be traced back to defined constellations of cell surface markers are functionally operational

subsets of T cells. However, this reviewer recognizes that careful functional characterization of these clusters (a minimum of 10 interesting clusters as defined in figure 3) is an enterprise exceeding the time needed to revise a manuscript. However, the authors ought to, at the very least, provide independent validation for the identity of these clusters. This is needed in view of the density of the information and the strict dependence on statistical treatment of the CyTOF data. Indeed, all the information presented in this manuscript depends on the processing of the CyTOF data by the t-SNE algorithm. Thus, the authors should demonstrate that the 10 clusters of CD8 T cells demonstrated in figure 3 exist and are identifiable by performing classical multicolor flow cytometry.

Here they should use panels of fluorescent antibody conjugates specific for the defining sets of cell surface markers in the multidimensional t-SNE maps.

Here the reviewer refers to the reliability of the CyTOF approach together with high dimensional reduction methods to disentangle different subsets of neoantigen-specific T cells. We addressed this concern and designed a flow cytometry panel that included markers that are characteristic for the individual clusters identified in untreated tumor bearing mice. We confirm that mLama4- and mAlg8-specific T cells represent a heterogeneous population and that different subsets can be identified by the signature markers that are prominent for each of the individual clusters (Page: 13) (Supplementary Figure 2). Moreover, by performing these stainings on tumors from individual mice we eliminate the possibility that the heterogeneity of neoantigen-specific T cells we identified before by mass cytometry results from the fact that several tumors had been pooled for the analysis. Although 10 clusters were delineated and described in our analysis, it was not our intention to conclude that there are indeed 10 functionally relevant subsets of antigen-specific T cells in these tumors. Instead, our intention was to broadly describe the composition of tumor-specific cells and this led to the finding that these cells are heterogeneous. We believe that our new data support this assertion. We have clarified in the text that the 10 different clusters are used as a method to describe our data and allow us to validate the finding that these cells are heterogeneous but should not imply that we think there are always exactly 10 different types of tumor-specific T cells in these tumor infiltrates (Page: 21).

4) Typos and minor point:

Page 7: line 15, there is some link missing, a parenthesis is also missing

Page 9: line 18, isotype (not isotope)

Page 10: line 6, increased proliferation (and not increases)

Page 16: line 14, the reference should follow the same format as in the rest of

the manuscript (numbered)

- We thank the reviewer and have corrected the errors accordingly. We apologize for not catching these errors before submission.

In the introduction, the authors should update the part on neoantigens in clinic by discussing the paper by McGranahan ... Swanton, Science 2016 (Clonal neoantigens elicit T cell immunoreactivity and sensitivity to immune checkpoint blockade).

- We agree with the reviewer and have added description of this important publication to the introduction (Page: 4).

Reviewer #2 (Remarks to the Author):

The authors combined mass cytometry analysis with multiplex combinatorial tetramer staining to identify and characterize neoantigen-specific CD8+ T cells across tissues in mice bearing T3 methylcholanthrene (MCA)-induced sarcomas following checkpoint blockade immunotherapy. One of the major conclusions that two neoantigen-specific CD8 cells were activated with anti-CTLA4 treatment was already reported in the previous report (reference 5). The further classification of neoantigen-specific T cells using high dimensional phenotypic profiling is very interesting, but very descriptive. In overall, this manuscript is well written, but they need to characterize the biological significance of these subpopulations.

(1) A simple question is whether TCR repertoires in these 10 clusters in Figure 3A are identical, similar, or different. This is very important to address whether the affinity of TCR to the HLA-antigen complex influences to the phenotype differences. Since the numbers of cells in some clusters are small, they should characterize TCR sequences of at least major clusters, for example, clusters 1-5 for mLama4 T cells and for clusters 1-6 for mAlg8.

- Although we entirely agree with this reviewer that the analysis of TCR sequences of these heterogeneous populations of these tumor-specific T cells could be very interesting, we think that this set of experiments are beyond the scope of this study. The purpose of this study was to illustrate the utility of for simultaneously identification and phenotypic profiling of tumor-specific T cells. To prove the utility of this approach we show that it can be useful in providing biological insight. In this respect, we have made several conclusions some of which have now been validated by follow-up experiments.
 - (i) neoantigen-specific TILs can constitute a heterogeneous population,
 - (ii) neoantigen-specific TILs can phenotypically and functionally be

different, and

(iii) neoantigen-specific TILs displaying different phenotypes can morph into novel phenotypic subsets following checkpoint blockade immunotherapy.

(2) They should also characterize the cytotoxic activity of T cells in some clusters to demonstrate the functional significance of these phenotypic differences. It is totally unclear in the difference in expression levels of cytotoxic molecules related to CD8 anti-tumor activity.

They must provide some biological aspects how the cells in individual clusters have different functional roles.

- We agree with the reviewer and have performed additional flow cytometry experiments to assess granzyme B production of the phenotypically different antigen-specific TIL populations that can be found in untreated tumor bearing mice. Interestingly, we detected that granzyme B expression is restricted to certain antigen-specific T cell clusters thus suggesting different anti-tumor activity by these cells. (Page: 14 and 21) (Figure 3C).

Reviewer #3 (Remarks to the Author):

1. The manuscript claims that checkpoint blockade reshapes the heterogeneity of intratumoral neo-antigen specific CD8+ T cells. This claim is too broad as it focuses on CTLA-4 blockade only, there is no change in the epitopes recognized and so far only phenotypic differences are demonstrated. A claim that CTLA-4 blockade changes the phenotype of tumor-specific CD8+ T cells in one mouse model is justified.
- We performed additional experiments with PD-1 blockade and observed similar changes to those induced by anti-CTLA-4 treatment (Page: 14) (Supplementary Figure 3). Our aim was to describe the heterogeneity of T cells targeting the same tumor-specific antigen. To our knowledge, this is the first study that shows such a degree of heterogeneity within a population of T cells harboring the same antigen specificity as well as the dramatic phenotypic alterations that are associated with checkpoint blockade immunotherapy. Although we did not detect novel antigen-specificities, we feel that the title chosen well reflects our findings. However, as recommended, we amended the title accordingly to clarify that these observations were made in mice only.
2. The authors use a recently published murine tumor model in which two neo-epitopes were identified. The results on page 6-8 describe a

screen on TIL, spleen and lymph nodes using combinatorial coded tetramers in which 79 new potential neo-epitopes as well as the two previously identified epitopes is shown. No new epitopes were identified. This seems to be expected as I can imagine that a similar type of screen has been performed to identify the known two epitopes (their ref 5). If CTLA-4 blocking would have changed this (which did not occur) than it would be justified to report this screen in the main figures but now it seems redundant and it could have been shown in one of the supplemental figures. A brief report that such a screen did not reveal any other neo-epitopes to be recognized, neither spontaneously nor after CTLA-4 blocking would have been sufficient. The method itself has been published before (their ref 23) and as such novelty is low for this part of the study.

- We respectfully disagree with the conclusion that the concept and findings presented in this paper are not new. Our previous work focused solely on TILs and 67 different epitope candidates for which had been screened using a fluorescence based double coding approach. This technique did not facilitate the possibility to simultaneously assess all of the predicted binders within the same sample nor does it allow to simultaneously screen for these epitopes across tissues. Since anti-CTLA-4 treatment has been reported to be able to affect T-cell priming (Reference #43), the inclusion of peripheral tissues is a reasonable rationale. Although the method itself has been published before, to our knowledge this is the first report that demonstrates value of using a mass cytometry based multiplexed combinatorial tetramer screening approach for the identification of neoantigen-specific T cells. The demonstration that this screen can be applied simultaneously across different tissues and clearly identifies neoantigen-specific T cell populations within these is particularly relevant for patient samples that are usually limited in size and where for instance only blood but not tumor samples can be collected.
3. The first results in this study reveal upregulated Tim-3 and PD-1 in TIL. This brings up the question why the authors chose to study the effects of CTLA-4 blocking rather than PD-1 (or Tim-3) blocking. It is of strong interest to know whether blocking these other molecules also results in a strong phenotypic change or whether the two antibodies may have complementary effects. Especially, since these CTLA-4 blocking and PD-1 blocking is tested simultaneously in the human setting. I would recommend to add such a study to the current one.
- The scope of our work was to demonstrate the capability of a mass cytometry based multiplexed tetramer staining approach to screen for a large number of neoantigen candidates across tissues while retaining the capacity to perform in depth profiling and subset identification of

neoantigen-specific T cells in a tumor model that is responsive to checkpoint blockade immunotherapy. To these ends we performed our experiments by using CTLA-4 as checkpoint inhibitor since blocking has been shown to result in remarkable changes in neoantigen-specific T cells in this model (anti-CTLA-4, Gubin et al., 2014) Nevertheless, we followed the reviewer's recommendation and performed additional experiments to assess the phenotypic changes of neoantigen-specific T cells in tumor bearing mice undergoing anti-PD-1 checkpoint blockade immunotherapy. Similarly to anti-CTLA-4 treatment we observed a dramatic phenotypic alteration in these cells when tumor bearing mice underwent anti-PD-1 immunotherapy (Page: 14 and 22) (Supplementary Figure 3).

4. The results described at page 10 (Figure 2) is the actual novel part of the study, which I would consider to be the start of the results section. At page 12 the authors state that elevated expression of the immune inhibitory receptors are widely accepted to describe an exhausted or dysfunctional T cell subset. This view, however, has been challenged by many who show that such cells may actually represent recently activated T cells (e.g. Tas et al Cancer Res 2016; Zelinkyy et al J. Immunol 2011; Gros et al. JCI 2014). Indicating that the expression of these markers do not indicate exhausted/dysfunctional T cells per sé. The authors should more carefully describe their results refraining from statements about functional activity as this has not been tested in this study.
 - We appreciate this reviewer's comments about the general view on the functional status of T cells expressing such immune inhibitory molecules. We therefore removed the general statement about exhaustion markers in tumors and refer to the preceding study, where it clearly has been demonstrated that in this model T cells expressing such molecules are accompanied by dysfunctional effector functions that are eliminated in response to checkpoint blockade (Page: 12).
5. At page 12, the authors conclude that without CTLA-4 blocking the neo-epitope specific CD8+ T cells are among clusters 1-6 (out of 10). This is based on a cut-off of 10% It is not clear why they used this cut-off. Scrutinizing the data reveals that the majority of mLAMA-4 specific T cells cluster in C1-C3 (>75%), while the majority of mAlg8 cluster in C3-6, C9 & C10 (>60%), based on the frequencies provided. So there is clear phenotypic heterogeneity between the two different antigen-specific CD8+ T cell populations. The authors should describe this much better rather than concluding that for both specificities they mostly fall into C1-C6. Their final conclusion on this matter (page 13) is correct.

- We set a 10% cut-off according to the cell background levels to avoid the selection of small clusters for a simplified way to describe the heterogeneity observed amongst tumor-specific TILs. This way we aimed to provide visual access to three highlights of this study, (i) neoantigen-specific TILs can constitute a heterogeneous population, (ii) neoantigen-specific TILs targeting different epitopes can phenotypically be different, and (iii) neoantigen-specific TILs displaying different phenotypes can morph into phenotypically similar subsets following checkpoint blockade immunotherapy. However, to better emphasize the phenotypic heterogeneity between the two different antigen-specific CD8+ T cell populations, we followed the reviewer's suggestion and amended the part accordingly: "In particular, whereas the majority of mLama4-specific T cells could be detected within C1-C3, we found the highest frequencies of mAlg8-specific T cells to be present in C3-C6" (Page: 12-13).

6. It is spectacular to see how CTLA-4 blocking drives the majority of the cells into C8-10 (rather than C7-10, as there is more difference in C7 between the two groups). It not unexpected as in their earlier paper (their ref 5) the RNAseq and GSEA data set analyses already showed that CTLA-4 blocking induced functional differences associated with T cell activation. Anyway, the data indicate that CTLA-4 blocking really converges the phenotype of the two different neo-epitope specific populations. To understand the relevance of these mouse model data is it is highly recommended to complement the data set showing that the T cells with the new phenotype (C8-C10), indeed bear more functionality than the T cells in the other clusters.

- In addition to the RNAseq and GSEA data our precedent study has also shown that anti-CTLA-4 treatment drives mLama4 and mAlg8-specific T cells towards a phenotype that displays lesser expression of inhibitory molecules (Tim-3, Lag-3, and PD-1) and further renders these cells more functionally active as seen by elevated production levels of cytotoxic molecules. Our study was not intended to recapitulate these findings and therefore we did not perform additional experiments that confirm the functionality of antigen-specific T cells found in the novel cluster composition appearing in response to anti-CTLA-4 therapy.

7. The authors also interrogated potential phenotypic changes in the periphery of treated mice. Based on the results presented in Figure 4b the authors conclude that the changes found are only seen in tetramer (neo-epitope specific) CD8+ T cells since tetramer negative T cells do not show significant changes. In my opinion one should be careful to

make such a statement as the population of tetramer-negative cells also contain a lot of naive T cells. These cells will dilute the signals provided by other memory or activated CD8+ T cells. If the authors want to make such a statement they should provide data on unrelated but previously activated T cells (e.g. virus specific T cells).

- We understand the reviewers concern about contaminating naïve cells that might influence the analysis of the effects of anti-CTLA-treatment on the tetramer negative population in the periphery. However, we would like to clarify that we did not assess the effects of checkpoint blockade immunotherapy on the tetramer negative fraction found in the periphery. For the analysis of the tetramer negative fraction in the tumor, we have shown that these cells are CD62L negative but CD44 high and thus do not display a naive phenotype (Figure 1 C). Nevertheless to eliminate these concerns and to further clarify this, we have amended this statement in the revised manuscript: “Due to low frequencies of antigen-specific T cells found in each of these peripheral tissues assessed and to allow for comparisons of the phenotypes of bulk CD8⁺ T cells we also included tetramer-negative CD8⁺ T cells from the corresponding compartment for this analysis” into: “Due to low frequencies of antigen-specific T cells found in each of these peripheral tissues assessed and to allow for comparisons of the phenotypes of bulk CD8⁺ T cells infiltrating the tumors we also included tetramer-negative CD8⁺ T cells for this analysis” (Page: 17).

7. In summary, the authors used one mouse model to provide good evidence that CTLA-4 blocking alters the phenotype of tumor-specific T cells, most prominently in the tumor itself. This is highly interesting but a number of questions that are important to address remain:

a) Are the changes observed coupled to a more functionally effective T cell?

- We have previously shown that checkpoint blockade immunotherapy results in an enhanced capacity of neoantigen-specific T cells carrying out effector functions. Checkpoint blockade immunotherapy rendered such T cells more activated resulting in T cell dependent tumor regression.

b) How general is this phenomenon, do the authors also observe this in a second mouse tumor model?

- We have not studied anti-CTLA-4 treatment in a different mouse model. Our observations are based on the effects of CTLA-4 blocking on tumor-specific T cells. To generalize this phenomenon it would be necessary to identify different mouse models with known tumor-specific antigens that are responsive to checkpoint blockade immunotherapy.

This, however, was not the intention of our study.

c) Does this change also occur in patients treated with CTLA-4?

- We have not extended this study to human patient material and therefore cannot make any conclusions about this. However, recent studies in cancer patients receiving anti-CTLA-4 or anti-PD-1 immunotherapy that we have cited in this study show that neoantigen-specific T cells increase in numbers (Reference #9 and #11) and cells derived from anti-PD-1 treated patients also displayed a polyfunctional phenotype after treatment (Reference #11). Since we observed phenotypic changes in tumor-specific T cells from either anti-CTLA-4 or anti-PD-1 treated animals, we would expect that CTLA-4 blocking also alters the phenotypes of tumor-specific T cells in humans. However, more evidence is required to determine whether similar changes as reported here can occur in patients in response to anti-CTLA-4 checkpoint blockade immunotherapy.

d) Is this effect specific for CTLA-4 blocking or does it also occur when other checkpoint blockers are used? Alternatively, would the changes induced by different blockers complement each other? Based on the RNAseq and GSEA data set provided in the earlier study (their ref 5) one should expect that.

- The reviewer brings up an important point. We therefore conducted additional experiments using PD-1 as checkpoint inhibitor and observed similar effects on the alteration of tumor-specific TILs as seen by anti-CTLA-4 treatment (Page: 15 and 22)(Supplementary Figure 3).

8. The discussion (pages 18-23) is rather lengthy and the first two pages more or less discuss the use of the combinatorial coded tetramers and the CyTOF as a tool to identify and phenotype tumor-specific T cells. As in this study no other epitopes were identified, and the method has already been discussed (their ref 23) this part is rather redundant and could be removed easily.

- We followed the reviewer's suggestions and shortened this part accordingly.

9. Furthermore, at page 20 the authors conclude that in their model CTLA-4 blocking did not lead to the appearance of novel T-cell specificities...suggesting that CTLA-4 acted only on pre-existing T cells. This can not be concluded as for none of the other peptides it has been shown that they function as actual epitopes. For such a statement, they should focus on other models for which it is known that epitopes are present but do not lead to a spontaneous immune response.

We appreciate this concern and we have rephrased this section to clarify about what we can and cannot conclude. Note, however that this sentence was phrased as a hypothetical interpretation of the data rather than a conclusion. We agree that we cannot conclude that there aren't other epitopes (besides the two dominant epitopes described) involved in the tumor rejection mechanism associated with anti-CTLA4 treatment. However, because we do see significant phenotypic changes in the two dominant with treatment and we do not observe any new epitopes (within the range candidates screened), we think that our data suggest but do not prove that T cells specific for these two dominant epitopes are involved in the rejection process.

Reviewer #4 (Remarks to the Author):

The manuscript by Fehlings et al describes the combined use of mass cytometry and multiplex combinatorial tetramer staining to identify and characterize neoantigen-specific CTLs in sarcoma bearing mice. They tested 81 candidate antigens and discovered T cells specific for two previously identified neo-antigens. They found tumor-infiltrating T cells to be heterogeneous. anti-CTLA-4 immunotherapy drastically changed their profile. This is a technically superbly executed set of experiments by the Newell lab. The paper is well-written and the conclusions are supported by the data they describe. There are only a few shortcomings which should be addressed:

- We thank the reviewer for the positive feedback and we are pleased that our experiments have been viewed as “technically superb”.

The paper is an excellent technical proof-of-principle report on the feasibility of the method. The combination of CyTOF and the tetramer library is really clever, but was already described previously in their Nat Biotech paper from 2013. Here, the only novel thing is the use of known cancer antigens (Ref. 5, cited 14 times) compared to the previously investigated viral antigens. This is however mitigated by the technical brilliance of the work, but it would have been perhaps a good idea to extend the study to identify new antigens (unknown) and thus use additional cancer cell lines (e.g. B16).

- We agree that this study proves feasibility for the use of a mass cytometry based multiplexed combinatorial tetramer staining approach for the identification of antigen-specific T cell. However, we respectfully disagree that the only novelty shown in the present study relies on the use of known cancer antigens that have been described before. In contrast to the previous work where a multiplexing approach was applied on human blood samples (Newell et al., 2013), here we demonstrate that this approach can be translated into the investigation of tumor-specific T cells from tumor tissues simultaneously with

peripheral tissues from the same group. To our knowledge this is the first study that carries out and validates the feasibility of such a comprehensive analysis. Moreover, by combining this approach with the t-SNE high dimensionality reduction tool we were able to reveal a high level of heterogeneity of antigen-specific TILs that has not been described in this extend before. By choosing this model we were able to demonstrate the feasibility of our method to detect neoantigen-specific T cells and to further deeply profile their phenotypic characteristic in the context of checkpoint blockade immunotherapy. Inclusion of another tumor model was not intended and was not the scope of our study.

The conclusions are based on the assumption that the t-SNE algorithm allows the definition of cell clusters. However, this algorithm is instead a visualization tool for dimensionality reduction, which per se does not cluster. The gating used in fig.2A, does not even allow a “visual segregation” of the different populations as depicted by the authors. For this reason, I recommend the use of automated-clustering algorithms (like self-organizing maps) to confirm the findings and characterize the different immune population in a more unbiased manner.

- In general, our purpose was to broadly describe the phenotypic profiles of tumor-specific T cells within tumors and lead us to develop novel hypotheses about the relevance of heterogeneity within these cell populations. These hypotheses were subsequently tested using standard gating approaches, as described. Although automated clustering algorithms can be useful, in this instance, we argue that manually delineated cell clusters (leveraging our human ability to interpret tSNE plots) allows us to more accurately delineate cell subsets. Nonetheless, to quantitatively address this important issue raised by the reviewer and to show that our manual cluster gating strategy is not entirely arbitrary or inaccurate, we performed automated clustering to validate our definitions of distinct cell clusters. To assess the consistency of the manual clusters' delineation with automated clustering, we performed k-means clustering of the t-SNE output, using 10 centers and 1000 random repeats. The chi-squared test was used to assess the correlation between the two grouping methods. Using this method we detected a similar clustering scheme. We feel that our manual clustering method is even more accurate by disentangling subtle differences between clusters 4,5, and 7 according to the heatmap presented in Figure 2A. The automated clustering data is presented below and discussed in the manuscript (Page: 21 and 30).

For Fig.1A is there biological control for the other tumor epitopes?

- Since all of the tumor epitopes assessed are potential candidates that result from the combination of different prediction algorithms we do not have controls for these epitopes. However, for negative control purposes we included the SIINFEKL epitope in some of our screens and validated the non-reactivity of T cells with those tumor epitopes. We have added this statement to our results section (Page: 7).

Fig.3: I think they over-interpreted the data. The effect of CTLA-4 must be compared with the overall impact on the tetramer negative subset. What happens when one compares the overall population of CD8 TILs profile before and after a CTLA-4 therapy? Most of the stratifying markers are also associated with an cell activation, and from this analysis one cannot dissect the direct effect of CTLA-4 on these subsets rather than an overall effect on the tumor microenvironment.

- Our approach allows us to specifically identify tumor-specific TILs and to directly assess the effects of anti-CTLA-4 treatment on these cells. We have investigated the effects of CTLA-4 blocking on the tetramer negative-cell fraction and did not observe remarkable changes in the expression of the marker molecules assessed (Fig. 4B). Although we detected some overlapping regions between tetramer negative and positive cells, these reflect a minor population of the overall tetramer negative population only. Treatment induced effects on these cells would not be remarkable deciphered in a global analysis of tetramer negative TILs.

In Fig.4B they compared the expression of some activation/maturation/exhaustion markers on neo-antigen specific CD8 T cells. Interestingly, within the tumor environment, the heatmap depicts a tetramer negative compartment of CD8 T cells that express very low levels of each of the analyzed markers, including CD44, CD27, CD5 and exhaustion markers such as PD1. What are those cells? On the same line, they do not provide the tetramer negative profile from spleen and lymph nodes. What is the impact of anti-CTLA-4 on these cells?

The aim of this study was to assess neoantigen-specific T cells from tumors and peripheral tissues. Since the majority of CD8+ T cells in the periphery are tumor unrelated we did not assess the phenotypes of bulk CD8+ T cells from the periphery. However, we feel it is an interesting fact that several T cells infiltrate the tumors that are not specific for any of the tumor-antigens tested. We don't know the role of these TILs and have discussed this in the manuscript with an emphasis for the need to study the role of such cells in the future (Page: 20). Indeed, as is pointed out, the heatmap on Fig. 4B shows that the average expression of these exhaustion-associated markers are expressed less by tetramer-negative cells. Z-score normalized values are used as a way to specifically compare the relative expression levels of each marker between treated and untreated mice and between the different antigen-specificities assessed. It is not possible to infer the absolute level of expression for each of these markers from this plot – instead the reviewer should refer to Figure 1 and other figures within this manuscript. For instance, in this case although CD44 expression is slightly less on tetramer-negative TILs, the majority of these cells do actually express CD44, as is illustrated in Fig. 1.

Lastly, the title is misleading. I expected that they applied this to patient material. The title should/must say “mouse”!

- We agree that the title could be misleading and amended this accordingly.

REVIEWER COMMENTS:

Previous round of revisions:

Reviewer #1 (Remarks to the Author):

M. Fehlings et al. use an approach combining mass cytometry and multiplexed multimer labeling to characterize neoantigen-specific CD8 T cell responses in different tissues of tumor bearing mice upon immune checkpoint blockade therapy. This is basically a methods paper illustrating the usefulness of this combined experimental fine grained analysis to generate high content information about the dynamics of T cell responses to predicted neoantigens in an experimental chemically induced tumor model.

The algorithm used to deconvolute high dimensional phenotypic data (t-SNE) was reported and illustrated back in 2013, in work from the GP Nolan's lab. In the present manuscript the t-SNE statistical approach is used to generate two-dimensional clusters of antigen-specific CD8 T cells. It shows a rather impressive power to reduce the complex cobweb of median cell surface marker expression values into subsets of apparently distinct populations of T cells which are shifted upon anti-CTLA-4 treatment. The results display is elegant and very useful as it points to a tractable way of capturing the complexity of the changes operated in the makeup of neoantigen-specific CD8 T cells during anti-CTLA-4 therapy, which in this model is highly clinically relevant.

The manuscript is clearly written and the results are worth sharing with the scientific community because of their key relevance for the very fast moving field of cancer immunotherapy with immune checkpoint blockade. In particular, these analyses significantly expand the initial set of observations published by Gubin M et al, Nature 2014. There are in this regard, a number of specific points that need to be addressed in order to better put the present results into context with the previous report on the targeting of two dominant neoantigens by immune checkpoint blockade therapy.

We thank this reviewer for the positive comments and are pleased that our study was so well received.

- 1) The RNAseq analyses of neoantigen-specific CD8 T cells (anti-mLama4) harvested from TILs after anti-CTLA-4 therapy showed a distinctive gene signature pointing to increased NFAT and JAK-STAT signaling, cellular proliferation/cell cycle and activation of effector T cells (Gubin et al. 2014). It is a bit disappointing that no proliferation/cell cycle markers were included in the panel design used in this manuscript so as to assess the extent to which neoantigen specific CD8 T cells are in cell cycle, in particular which in the S-phase, by the time of analysis in the isotype and

anti-CTLA-4 groups of tumor bearing mice. Inclusion of this analysis would increase the potential impact of this manuscript.

To address this important issue raised by the reviewer, we designed a flow cytometry panel including Ki-67 to assess the proliferation status of distinct neoantigen-specific TIL subsets (Figure 3C). When we analyzed the proliferation status from different T-cell clusters found in untreated tumor bearing mice, we detected that approximately half of the cells found in the clusters were positive for Ki-67. However, regardless of clear phenotypic differences amongst the T-cell clusters we did not detect remarkable differences in the proliferation levels of the cells within these different clusters (Page: 14).

- 2) In the original work published by Gubin et al, the tumor model also responded to anti-PD-1 and to the anti-CTLA-4/anti-PD-1 monotherapy and combination therapy, respectively. The authors should also show the phenotypic changes that may be detected using this powerful analytic approach upon anti-PD-1 therapy. In fact, the latter therapy is nowadays far more relevant to the current clinical situation as anti-PD-1 therapy is showing activity in a surprisingly large number of different tumor types.

The scope of our work was to demonstrate the feasibility of a mass cytometry based multiplexed tetramer staining approach to screen for a large number of neoantigen candidates across tissues while retaining the capacity to perform in depth profiling and subset identification of neoantigen-specific cells in a tumor model that is responsive to checkpoint blockade immunotherapy. To these ends we performed our experiments by using one checkpoint inhibitor that has been shown to result in remarkable changes in this model (anti-CTLA-4, Gubin et al., 2014). Nevertheless, we agree with the reviewer on the importance of assessing tumor antigen-specific T cells upon anti-PD-1 immunotherapy and performed additional experiments with mice undergoing anti-PD-1 treatment and assessed the phenotypic changes on mLama4-neoantigen-specific TILs and their peripheral counterparts. We found that anti-PD-1 treatment induced substantial phenotypical changes on mLama4-specific T cells in the tumors and that these changes are comparable to those observed with anti-CTLA-4 treatment. Notably, we also found that these changes are restricted to TILs, since we did not detect such changes when we investigated mLama4-specific T cells in draining lymph nodes following anti-PD1 immunotherapy. (Page: 15 and 22) (Supplementary Figure 3).

- 3) The most important concern with the present manuscript is the extent to which the clusters of T cells which can be traced back to defined

constellations of cell surface markers are functionally operational subsets of T cells. However, this reviewer recognizes that careful functional characterization of these clusters (a minimum of 10 interesting clusters as defined in figure 3) is an enterprise exceeding the time needed to revise a manuscript. However, the authors ought to, at the very least, provide independent validation for the identity of these clusters. This is needed in view of the density of the information and the strict dependence on statistical treatment of the CyTOF data. Indeed, all the information presented in this manuscript depends on the processing of the CyTOF data by the t-SNE algorithm. Thus, the authors should demonstrate that the 10 clusters of CD8 T cells demonstrated in figure 3 exist and are identifiable by performing classical multicolor flow cytometry.

Here they should use panels of fluorescent antibody conjugates specific for the defining sets of cell surface markers in the multidimensional t-SNE maps.

Here the reviewer refers to the reliability of the CyTOF approach together with high dimensional reduction methods to disentangle different subsets of neoantigen-specific T cells. We addressed this concern and designed a flow cytometry panel that included markers that are characteristic for the individual clusters identified in untreated tumor bearing mice. We confirm that mLama4- and mAlg8-specific T cells represent a heterogeneous population and that different subsets can be identified by the signature markers that are prominent for each of the individual clusters (Page: 13) (Supplementary Figure 2). Moreover, by performing these stainings on tumors from individual mice we eliminate the possibility that the heterogeneity of neoantigen-specific T cells we identified before by mass cytometry results from the fact that several tumors had been pooled for the analysis. Although 10 clusters were delineated and described in our analysis, it was not our intention to conclude that there are indeed 10 functionally relevant subsets of antigen-specific T cells in these tumors. Instead, our intention was to broadly describe the composition of tumor-specific cells and this led to the finding that these cells are heterogeneous. We believe that our new data support this assertion. We have clarified in the text that the 10 different clusters are used as a method to describe our data and allow us to validate the finding that these cells are heterogeneous but should not imply that we think there are always exactly 10 different types of tumor-specific T cells in these tumor infiltrates (Page: 21).

4) Typos and minor point:

Page 7: line 15, there is some link missing, a parenthesis is also missing

Page 9: line 18, isotype (not isotope)

Page 10: line 6, increased proliferation (and not increases)

Page 16: line 14, the reference should follow the same format as in the rest of the manuscript (numbered)

We thank the reviewer and have corrected the errors accordingly. We apologize for not catching these errors before submission.

In the introduction, the authors should update the part on neoantigens in clinic by discussing the paper by McGranahan ... Swanton, Science 2016 (Clonal neoantigens elicit T cell immunoreactivity and sensitivity to immune checkpoint blockade).

We agree with the reviewer and have added description of this important publication to the introduction (Page: 4).

Reviewer #2 (Remarks to the Author):

The authors combined mass cytometry analysis with multiplex combinatorial tetramer staining to identify and characterize neoantigen-specific CD8+ T cells across tissues in mice bearing T3 methylcholanthrene (MCA)-induced sarcomas following checkpoint blockade immunotherapy. One of the major conclusions that two neoantigen-specific CD8 cells were activated with anti-CTLA4 treatment was already reported in the previous report (reference 5). The further classification of neoantigen-specific T cells using high dimensional phenotypic profiling is very interesting, but very descriptive. In overall, this manuscript is well written, but they need to characterize the biological significance of these subpopulations.

(1) A simple question is whether TCR repertoires in these 10 clusters in Figure 3A are identical, similar, or different. This is very important to address whether the affinity of TCR to the HLA-antigen complex influences to the phenotype differences. Since the numbers of cells in some clusters are small, they should characterize TCR sequences of at least major clusters, for example, clusters 1-5 for mLama4 T cells and for clusters 1-6 for mAlg8.

Although we entirely agree with this reviewer that the analysis of TCR sequences of these heterogeneous populations of these tumor-specific T cells could be very interesting, we think that this set of experiments are beyond the scope of this study. The purpose of this study was to illustrate the utility of for simultaneously identification and phenotypic profiling of tumor-specific T cells. To prove the utility of this approach we show that it can be useful in providing biological insight. In this respect, we have made several conclusions some of which have now been validated by follow-up experiments.

(i) neoantigen-specific TILs can constitute a heterogeneous population, (ii)

neoantigen-specific TILs can phenotypically and functionally be different, and (iii) neoantigen-specific TILs displaying different phenotypes can morph into novel phenotypic subsets following checkpoint blockade immunotherapy.

(2) They should also characterize the cytotoxic activity of T cells in some clusters to demonstrate the functional significance of these phenotypic differences. It is totally unclear in the difference in expression levels of cytotoxic molecules related to CD8 anti-tumor activity.

They must provide some biological aspects how the cells in individual clusters have different functional roles.

We agree with the reviewer and have performed additional flow cytometry experiments to assess granzyme B production of the phenotypically different antigen-specific TIL populations that can be found in untreated tumor bearing mice. Interestingly, we detected that granzyme B expression is restricted to certain antigen-specific T cell clusters thus suggesting different anti-tumor activity by these cells. (Page: 14 and 21) (Figure 3C).

Reviewer #3 (Remarks to the Author):

1. The manuscript claims that checkpoint blockade reshapes the heterogeneity of intratumoral neo-antigen specific CD8+ T cells. This claim is too broad as it focuses on CTLA-4 blockade only, there is no change in the epitopes recognized and so far only phenotypic differences are demonstrated. A claim that CTLA-4 blockade changes the phenotype of tumor-specific CD8+ T cells in one mouse model is justified.

We performed additional experiments with PD-1 blockade and observed similar changes to those induced by anti-CTLA-4 treatment (Page: 14) (Supplementary Figure 3). Our aim was to describe the heterogeneity of T cells targeting the same tumor-specific antigen. To our knowledge, this is the first study that shows such a degree of heterogeneity within a population of T cells harboring the same antigen specificity as well as the dramatic phenotypic alterations that are associated with checkpoint blockade immunotherapy. Although we did not detect novel antigen-specificities, we feel that the title chosen well reflects our findings. However, as recommended, we amended the title accordingly to clarify that these observations were made in mice only.

2. The authors use a recently published murine tumor model in which two neo-epitopes were identified. The results on page 6-8 describe a screen

on TIL, spleen and lymph nodes using combinatorial coded tetramers in which 79 new potential neo-epitopes as well as the two previously identified epitopes is shown. No new epitopes were identified. This seems to be expected as I can imagine that a similar type of screen has been performed to identify the known two epitopes (their ref 5). If CTLA-4 blocking would have changed this (which did not occur) than it would be justified to report this screen in the main figures but now it seems redundant and it could have been shown in one of the supplemental figures. A brief report that such a screen did not reveal any other neo-epitopes to be recognized, neither spontaneously nor after CTLA-4 blocking would have been sufficient. The method itself has been published before (their ref 23) and as such novelty is low for this part of the study.

We respectfully disagree with the conclusion that the concept and findings presented in this paper are not new. Our previous work focused solely on TILs and 67 different epitope candidates for which had been screened using a fluorescence based double coding approach. This technique did not facilitate the possibility to simultaneously assess all of the predicted binders within the same sample nor does it allow to simultaneously screen for these epitopes across tissues. Since anti-CTLA-4 treatment has been reported to be able to affect T-cell priming (Reference #43), the inclusion of peripheral tissues is a reasonable rationale. Although the method itself has been published before, to our knowledge this is the first report that demonstrates value of using a mass cytometry based multiplexed combinatorial tetramer screening approach for the identification of neoantigen-specific T cells. The demonstration that this screen can be applied simultaneously across different tissues and clearly identifies neoantigen-specific T cell populations within these is particularly relevant for patient samples that are usually limited in size and where for instance only blood but not tumor samples can be collected.

3. The first results in this study reveal upregulated Tim-3 and PD-1 in TIL. This brings up the question why the authors chose to study the effects of CTLA-4 blocking rather than PD-1 (or Tim-3) blocking. It is of strong interest to know whether blocking these other molecules also results in a strong phenotypic change or whether the two antibodies may have complementary effects. Especially, since these CTLA-4 blocking and PD-1 blocking is tested simultaneously in the human setting. I would recommend to add such a study to the current one.

The scope of our work was to demonstrate the capability of a mass cytometry based multiplexed tetramer staining approach to screen for a large number of neoantigen candidates across tissues while retaining the capacity to perform in depth profiling and subset identification of

neoantigen-specific T cells in a tumor model that is responsive to checkpoint blockade immunotherapy. To these ends we performed our experiments by using CTLA-4 as checkpoint inhibitor since blocking has been shown to result in remarkable changes in neoantigen-specific T cells in this model (anti-CTLA-4, Gubin et al., 2014) Nevertheless, we followed the reviewer's recommendation and performed additional experiments to assess the phenotypic changes of neoantigen-specific T cells in tumor bearing mice undergoing anti-PD-1 checkpoint blockade immunotherapy. Similarly to anti-CTLA-4 treatment we observed a dramatic phenotypic alteration in these cells when tumor bearing mice underwent anti-PD-1 immunotherapy (Page: 14 and 22) (Supplementary Figure 3).

4. The results described at page 10 (Figure 2) is the actual novel part of the study, which I would consider to be the start of the results section. At page 12 the authors state that elevated expression of the immune inhibitory receptors are widely accepted to describe an exhausted or dysfunctional T cell subset. This view, however, has been challenged by many who show that such cells may actually represent recently activated T cells (e.g. Tas et al Cancer Res 2016; Zelinkyy et al J. Immunol 2011; Gros et al. JCI 2014). Indicating that the expression of these markers do not indicate exhausted/dysfunctional T cells per sé. The authors should more carefully describe their results refraining from statements about functional activity as this has not been tested in this study.

We appreciate this reviewer's comments about the general view on the functional status of T cells expressing such immune inhibitory molecules. We therefore removed the general statement about exhaustion markers in tumors and refer to the preceding study, where it clearly has been demonstrated that in this model T cells expressing such molecules are accompanied by dysfunctional effector functions that are eliminated in response to checkpoint blockade (Page: 12).

5. At page 12, the authors conclude that without CTLA-4 blocking the neo-epitope specific CD8+ T cells are among clusters 1-6 (out of 10). This is based on a cut-off of 10% It is not clear why they used this cut-off. Scrutinizing the data reveals that the majority of mLAMA-4 specific T cells cluster in C1-C3 (>75%), while the majority of mAlg8 cluster in C3-6, C9 & C10 (>60%), based on the frequencies provided. So there is clear phenotypic heterogeneity between the two different antigen-specific CD8+ T cell populations. The authors should describe this much better rather than concluding that for both specificities they mostly fall into C1-C6. Their final conclusion on this matter (page 13) is correct.

We set a 10% cut-off according to the cell background levels to avoid the

selection of small clusters for a simplified way to describe the heterogeneity observed amongst tumor-specific TILs. This way we aimed to provide visual access to three highlights of this study, (i) neoantigen-specific TILs can constitute a heterogeneous population, (ii) neoantigen-specific TILs targeting different epitopes can phenotypically be different, and (iii) neoantigen-specific TILs displaying different phenotypes can morph into phenotypically similar subsets following checkpoint blockade immunotherapy. However, to better emphasize the phenotypic heterogeneity between the two different antigen-specific CD8+ T cell populations, we followed the reviewer's suggestion and amended the part accordingly: "In particular, whereas the majority of mLama4-specific T cells could be detected within C1-C3, we found the highest frequencies of mAlg8-specific T cells to be present in C3-C6" (Page: 12-13).

6. It is spectacular to see how CTLA-4 blocking drives the majority of the cells into C8-10 (rather than C7-10, as there is more difference in C7 between the two groups). It not unexpected as in their earlier paper (their ref 5) the RNAseq and GSEA data set analyses already showed that CTLA-4 blocking induced functional differences associated with T cell activation. Anyway, the data indicate that CTLA-4 blocking really converges the phenotype of the two different neo-epitope specific populations. To understand the relevance of these mouse model data is it is highly recommended to complement the data set showing that the T cells with the new phenotype (C8-C10), indeed bear more functionality than the T cells in the other clusters.

In addition to the RNAseq and GSEA data our precedent study has also shown that anti-CTLA-4 treatment drives mLama4 and mAlg8-specific T cells towards a phenotype that displays lesser expression of inhibitory molecules (Tim-3, Lag-3, and PD-1) and further renders these cells more functionally active as seen by elevated production levels of cytotoxic molecules. Our study was not intended to recapitulate these findings and therefore we did not perform additional experiments that confirm the functionality of antigen-specific T cells found in the novel cluster composition appearing in response to anti-CTLA-4 therapy.

7. The authors also interrogated potential phenotypic changes in the periphery of treated mice. Based on the results presented in Figure 4b the authors conclude that the changes found are only seen in tetramer (neo-epitope specific) CD8+ T cells since tetramer negative T cells do not show significant changes. In my opinion one should be careful to make such a statement as the population of tetramer-negative cells also contain a lot of

naive T cells. These cells will dilute the signals provided by other memory or activated CD8+ T cells. If the authors want to make such a statement they should provide data on unrelated but previously activated T cells (e.g. virus specific T cells).

We understand the reviewers concern about contaminating naïve cells that might influence the analysis of the effects of anti-CTLA-treatment on the tetramer negative population in the periphery. However, we would like to clarify that we did not assess the effects of checkpoint blockade immunotherapy on the tetramer negative fraction found in the periphery. For the analysis of the tetramer negative fraction in the tumor, we have shown that these cells are CD62L negative but CD44 high and thus do not display a naive phenotype (Figure 1 C). Nevertheless to eliminate these concerns and to further clarify this, we have amended this statement in the revised manuscript: “Due to low frequencies of antigen-specific T cells found in each of these peripheral tissues assessed and to allow for comparisons of the phenotypes of bulk CD8⁺ T cells we also included tetramer-negative CD8⁺ T cells from the corresponding compartment for this analysis” into: “Due to low frequencies of antigen-specific T cells found in each of these peripheral tissues assessed and to allow for comparisons of the phenotypes of bulk CD8⁺ T cells infiltrating the tumors we also included tetramer-negative CD8⁺ T cells for this analysis” (Page: 17).

7. In summary, the authors used one mouse model to provide good evidence that CTLA-4 blocking alters the phenotype of tumor-specific T cells, most prominently in the tumor itself. This is highly interesting but a number of questions that are important to address remain:

a) Are the changes observed coupled to a more functionally effective T cell?

We have previously shown that checkpoint blockade immunotherapy results in an enhanced capacity of neoantigen-specific T cells carrying out effector functions. Checkpoint blockade immunotherapy rendered such T cells more activated resulting in T cell dependent tumor regression.

b) How general is this phenomenon, do the authors also observe this in a second mouse tumor model?

We have not studied anti-CTLA-4 treatment in a different mouse model. Our observations are based on the effects of CTLA-4 blocking on tumor-specific T cells. To generalize this phenomenon it would be necessary to identify different mouse models with known tumor-specific antigens that are responsive to checkpoint blockade immunotherapy. This, however, was not the intention of our

study.

c) Does this change also occur in patients treated with CTLA-4?

We have not extended this study to human patient material and therefore cannot make any conclusions about this. However, recent studies in cancer patients receiving anti-CTLA-4 or anti-PD-1 immunotherapy that we have cited in this study show that neoantigen-specific T cells increase in numbers (Reference #9 and #11) and cells derived from anti-PD-1 treated patients also displayed a polyfunctional phenotype after treatment (Reference #11). Since we observed phenotypic changes in tumor-specific T cells from either anti-CTLA-4 or anti-PD-1 treated animals, we would expect that CTLA-4 blocking also alters the phenotypes of tumor-specific T cells in humans. However, more evidence is required to determine whether similar changes as reported here can occur in patients in response to anti-CTLA-4 checkpoint blockade immunotherapy.

d) Is this effect specific for CTLA-4 blocking or does it also occur when other checkpoint blockers are used? Alternatively, would the changes induced by different blockers complement each other? Based on the RNAseq and GSEA data set provided in the earlier study (their ref 5) one should expect that.

The reviewer brings up an important point. We therefore conducted additional experiments using PD-1 as checkpoint inhibitor and observed similar effects on the alteration of tumor-specific TILs as seen by anti-CTLA-4 treatment (Page: 15 and 22) (Supplementary Figure 3).

8. The discussion (pages 18-23) is rather lengthy and the first two pages more or less discuss the use of the combinatorial coded tetramers and the CyTOF as a tool to identify and phenotype tumor-specific T cells. As in this study no other epitopes were identified, and the method has already been discussed (their ref 23) this part is rather redundant and could be removed easily.

We followed the reviewer's suggestions and shortened this part accordingly.

9. Furthermore, at page 20 the authors conclude that in their model CTLA-4 blocking did not lead to the appearance of novel T-cell specificities...suggesting that CTLA-4 acted only on pre-existing T cells. This can not be concluded as for none of the other peptides it has been shown that they function as actual epitopes. For such a statement, they should focus on other models for which it is known that epitopes are present but do not lead to a spontaneous immune response.

We appreciate this concern and we have rephrased this section to clarify about what we can and cannot conclude. Note, however that this sentence was

phrased as a hypothetical interpretation of the data rather than a conclusion. We agree that we cannot conclude that there aren't other epitopes (besides the two dominant epitopes described) involved in the tumor rejection mechanism associated with anti-CTLA4 treatment. However, because we do see significant phenotypic changes in the two dominant with treatment and we do not observe any new epitopes (within the range candidates screened), we think that our data suggest but do not prove that T cells specific for these two dominant epitopes are involved in the rejection process.

Reviewer #4 (Remarks to the Author):

The manuscript by Fehlings et al describes the combined use of mass cytometry and multiplex combinatorial tetramer staining to identify and characterize neoantigen-specific CTLs in sarcoma bearing mice. They tested 81 candidate antigens and discovered T cells specific for two previously identified neo-antigens. They found tumor-infiltrating T cells to be heterogeneous. anti-CTLA-4 immunotherapy drastically changed their profile.

This is a technically superbly executed set of experiments by the Newell lab. The paper is well-written and the conclusions are supported by the data they describe. There are only a few shortcomings which should be addressed:

We thank the reviewer for the positive feedback and we are pleased that our experiments have been viewed as “technically superb”.

The paper is an excellent technical proof-of-principle report on the feasibility of the method. The combination of CyTOF and the tetramer library is really clever, but was already described previously in their Nat Biotech paper from 2013. Here, the only novel thing is the use of known cancer antigens (Ref. 5, cited 14 times) compared to the previously investigated viral antigens. This is however mitigated by the technical brilliance of the work, but it would have been perhaps a good idea to extend the study to identify new antigens (unknown) and thus use additional cancer cell lines (e.g. B16).

We agree that this study proves feasibility for the use of a mass cytometry based multiplexed combinatorial tetramer staining approach for the identification of antigen-specific T cell. However, we respectfully disagree that the only novelty shown in the present study relies on the use of known cancer antigens that have been described before. In contrast to the previous work where a multiplexing approach was applied on human blood samples (Newell et al., 2013), here we demonstrate that this approach can be translated into the investigation of tumor-specific T cells from tumor tissues simultaneously with peripheral tissues from the same group. To our knowledge this is the first study that carries out and validates the feasibility of such a comprehensive analysis. Moreover, by

combining this approach with the t-SNE high dimensionality reduction tool we were able to reveal a high level of heterogeneity of antigen-specific TILs that has not been described in this extent before. By choosing this model we were able to demonstrate the feasibility of our method to detect neoantigen-specific T cells and to further deeply profile their phenotypic characteristic in the context of checkpoint blockade immunotherapy. Inclusion of another tumor model was not intended and was not the scope of our study.

The conclusions are based on the assumption that the t-SNE algorithm allows the definition of cell clusters. However, this algorithm is instead a visualization tool for dimensionality reduction, which per se does not cluster. The gating used in fig.2A, does not even allow a “visual segregation” of the different populations as depicted by the authors. For this reason, I recommend the use of automated-clustering algorithms (like self-organizing maps) to confirm the findings and characterize the different immune population in a more unbiased manner.

In general, our purpose was to broadly describe the phenotypic profiles of tumor-specific T cells within tumors and lead us to develop novel hypotheses about the relevance of heterogeneity within these cell populations. These hypotheses were subsequently tested using standard gating approaches, as described. Although automated clustering algorithms can be useful, in this instance, we argue that manually delineated cell clusters (leveraging our human ability to interpret tSNE plots) allows us to more accurately delineate cell subsets. Nonetheless, to quantitatively address this important issue raised by the reviewer and to show that our manual cluster gating strategy is not entirely arbitrary or inaccurate, we performed automated clustering to validate our definitions of distinct cell clusters. To assess the consistency of the manual clusters' delineation with automated clustering, we performed k-means clustering of the t-SNE output, using 10 centers and 1000 random repeats. The chi-squared test was used to assess the correlation between the two grouping methods. Using this method we detected a similar clustering scheme. We feel that our manual clustering method is even more accurate by disentangling subtle differences between clusters 4,5, and 7 according to the heatmap presented in Figure 2A. The automated clustering data is presented below and discussed in the manuscript (Page: 21 and 30).

For Fig.1A is there biological control for the other tumor epitopes?

Since all of the tumor epitopes assessed are potential candidates that result from the combination of different prediction algorithms we do not have controls for these epitopes. However, for negative control purposes we included the SIINFEKL epitope in some of our screens and validated the non-reactivity of T cells with those tumor epitopes. We have added this statement to our results section (Page: 7).

Fig.3: I think they over-interpreted the data. The effect of CTLA-4 must be compared with the overall impact on the tetramer negative subset. What happens when one compares the overall population of CD8 TILs profile before and after a CTLA-4 therapy? Most of the stratifying markers are also associated with an cell activation, and from this analysis one cannot dissect the direct effect of CTLA-4 on these subsets rather than an overall effect on the tumor microenvironment.

Our approach allows us to specifically identify tumor-specific TILs and to directly assess the effects of anti-CTLA-4 treatment on these cells.

We have investigated the effects of CTLA-4 blocking on the tetramer negative-cell fraction and did not observe remarkable changes in the expression of the marker molecules assessed (Fig. 4B). Although we detected some overlapping regions between tetramer negative and positive cells, these reflect a minor population of the overall tetramer negative population only. Treatment induced effects on these cells would not be remarkable deciphered in a global analysis of tetramer negative TILs.

In Fig.4B they compared the expression of some activation/maturation/exhaustion markers on neo-antigen specific CD8 T cells. Interestingly, within the tumor environment, the heatmap depicts a tetramer negative compartment of CD8 T cells that express very low levels of each of the analyzed markers, including CD44, CD27, CD5 and exhaustion markers such as PD1. What are those cells? On the same line, they do not provide the tetramer negative profile from spleen and lymph nodes. What is the impact of anti-CTLA-4 on these cells?

The aim of this study was to assess neoantigen-specific T cells from tumors and peripheral tissues. Since the majority of CD8+ T cells in the periphery are tumor unrelated we did not assess the phenotypes of bulk CD8+ T cells from the periphery. However, we feel it is an interesting fact that several T cells infiltrate the tumors that are not specific for any of the tumor-antigens tested. We don't know the role of these TILs and have discussed this in the manuscript with an emphasis for the need to study the role of such cells in the future (Page: 20). Indeed, as is pointed out, the heatmap on Fig. 4B shows that the average expression of these exhaustion-associated markers are expressed less by tetramer-negative cells. Z-score normalized values are used as a way to specifically compare the relative expression levels of each marker between treated and untreated mice and between the different antigen-specificities assessed. It is not possible to infer the absolute level of expression for each of these markers from this plot – instead the reviewer should refer to Figure 1 and other figures within this manuscript. For instance, in this case although CD44 expression is slightly less on tetramer-negative TILs, the majority of these cells do actually express CD44, as is illustrated in Fig. 1.

Lastly, the title is misleading. I expected that they applied this to patient material. The title should/must say “mouse”!

We agree that the title could be misleading and amended this accordingly.

Most recent round of revisions:

Reviewer#1

The authors have carefully revised their manuscript and added essential new data, including the extensive phenotypic characterization of neoantigen-specific T cells before and after PD-1 treatment. There are still specific concerns with the new data sets. While they show the Ki65 expression in TILs from untreated mice, the question was also concerning the relative levels of this proliferation marker after immune checkpoint blockade treatment. The pretreatment levels appear already quite high. Are they further increased upon anti-CTLA-4 and/or anti-PD-1 treatment?

We highly appreciated the suggestion of this reviewer to interrogate proliferation of the distinct antigen-specific T-cell clusters and agree that it was an important aspect when assessing heterogeneity of tumor-specific T cells that target different antigens.

We did not address Ki-67 expression in antigen-specific T cells after checkpoint blockade immunotherapy for the following reasons: (i) T cells for both antigen-specificities acquired a similar phenotype after treatment with a much lesser degree of heterogeneity, (ii) tumors from treated animals showed higher percentages and numbers of neoantigen-specific T cells compared to control mAbs treated mice (as shown here and in Gubin et al., 2014) and (iii) we have previously shown that treatment with anti-CTLA-4 increased the cellular proliferation/cell cycle of these cells (Gubin et al., 2014).

The other concern is relative to the legibility of the results depicted in supplementary figure 3A. It is unclear how the two dimensional plots reflect the changes upon treatment. The figure legend should better explain the color codes and what data is before and after treatment.

We agree with this concern and thank the reviewer for bringing this up. We have updated the figure as well as the figure legend accordingly.

Reviewer#2

Although they improved the manuscript extensively, they did not perform the TCR analysis. It is very important to know the biological differences such as T cell clonality or granzyme levels in these clusters. Since TCR sequencing methods are established now, I wish the authors add these analyses.

We would like to emphasize again that the purpose of our study was to illustrate the utility of a mass cytometry based multiplexed tetramer staining approach for the simultaneously identification and phenotypic profiling of tumor-specific T cells.

Nonetheless, we entirely agree with the reviewer that it would be very interesting to know the biological differences between the different tumor-specific T cell

clusters observed. Although TCR sequencing methods are established it still remains challenging due to the availability of sufficient cell numbers within the clusters. Thus, we anticipate that these data may not be simple to interpret given that TCR sequences can be very diverse even when specific for a single antigen. We therefore followed the alternative suggestion of this reviewer and determined granzyme B levels across the antigen-specific T cell clusters to evaluate functional differences amongst the cells found within these clusters.

Reviewer#3

The authors have elaborately addressed my previous remarks, most of them to my satisfaction. A few concerns still remain.

1. The title still does not properly reflect the findings. As there were no changes in specificities but only phenotypical changes, the title should indicate this. This is simple by inserting the word "phenotypical" between "...high-dimensional" and "heterogeneity...".

We thank the reviewer for this comment and agree that this amendment in the title now better clarifies the findings of our study.

2. I did not state that the concept and findings in this paper are not new. This remark was placed in the context of using mass cytometry for the combinatorial metal-based MHC tetramer screening in combination with other phenotypic markers. This work is well appreciated but has been demonstrated for 109 different tetramers in blood by the senior author (Newell et al. Nat Biotech 2013). Hence, the technique as well as the fact that it can be used for blood analyses is not new. Although the authors point out that this is the first report demonstrating the value of using mass cytometry for neo-antigen-specific T cells, I would consider this a stretch. There would for sure be a value when the authors had demonstrated that during therapy there were changes in the neo-antigens recognized. This was not the case. The sheer fact that now MHC tetramers with neo-antigen peptides are used should not be considered as something novel. Therefore, again I suggest to compress the description of the first part of the results section (pages 6-8).

We understand the reviewer's point that the technique applied here has been reported previously and that the use of neoantigens alone does not state something novel. However, here, we report for the first time that this approach can also efficiently be used for a comprehensive screen to simultaneously detect, profile, and compare neoantigen-specific T cells in primary tumor tissue and peripheral compartments. We feel that the possibility to perform such a comprehensive analysis is something new and demonstrates value of this approach. Nevertheless, we followed the reviewer's suggestion and shortened this part accordingly.

3. At this point only one mouse model has been used. The authors stated that to generalize this phenomenon they would require another mouse model with

known tumor-specific antigens and responsive to check-point blockade, but that it was not the intention of this study to identify such mouse model. This is a bit of a surprise as the authors must be aware of the MC38 mouse model which is responsive to the combination a-LAG3 Ab/a-PD1 Ab (Woo et al Cancer Res 2012, p917) and presents well-described neoepitopes to T cells (Yadav et al., Nature 2014, p572)

We are well aware of the existence of other cancer mouse models including the MC38 colon carcinoma that facilitate studying of neoantigen-specific T cells. We note, however that the antigens described in Yadav et al., do not actually represent endogenously targeted neo-antigen – rather they identified antigens that can be used effectively in a vaccine setting. Whatever the case, we think it would go well beyond the scope of this study to include an additional study on MC38 tumor bearing mice. In general, the comparison of the effects of checkpoint blockade immunotherapy on neoantigen-specific T cell across different tumor models, however, was not intended by our study. Moreover, addressing the question by the reviewer, whether the observed effects of anti-CTLA-4 cancer immunotherapy on neoantigen-specific cells could be generalized, another model with known tumor antigens would be necessary that at least (i) follows similar kinetics and (ii) responds to a single anti-CTLA-4 immunotherapy with efficient tumor regression. We like to emphasize again that this was beyond the scope of our study.

Reviewer#4
satisfied! well done!